# High throughput intracellular delivery by viscoelastic mechanoporation

Derin Sevenler [1] & Mehmet Toner[1,2] ✉

Brief pulses of electric field (electroporation) and/or tensile stress (mechanoporation) have been used to reversibly permeabilize the plasma membrane of mammalian cells and deliver materials to the cytosol. However, electroporation can be harmful to cells, while efficient mechanoporation strategies have not been scalable due to the use of narrow constrictions or needles which are susceptible to clogging. Here we report a high throughput approach to mechanoporation in which the plasma membrane is stretched and reversibly permeabilized by viscoelastic fluid forces within a microfluidic chip without surface contact. Biomolecules are delivered directly to the cytosol within seconds at a throughput exceeding 250 million cells per minute. Viscoelastic mechanoporation is compatible with a variety of biomolecules including proteins, RNA, and CRISPR-Cas9 ribonucleoprotein complexes, as well as a range of cell types including HEK293T cells and primary T cells. Altogether, viscoelastic mechanoporation appears feasible for contact-free permeabilization and delivery of biomolecules to mammalian cells ex vivo.

Emerging cell and gene therapies promise to revolutionize oncology and other fields but face unprecedented challenges in manufacturing. One obstacle has been the cost and complexity associated with efficiently delivering genetic materials and/or gene editing systems into very large numbers of cells ex vivo (hundreds of millions, up to trillions of cells for some applications)[1–3]. Broadly, there are three major ex vivo delivery methods currently in use for clinical-scale production: viral vectors, synthetic vectors, and electroporation. Viral vectors such as lentivirus are a well-established platform technology that is clinically approved for ex vivo genetic manipulation of immune cells, however they are unable to target a specific genetic locus, have a limited genetic payload, and can be expensive to produce in large quantities[4–6]. Synthetic vector systems such as DNA-cationic polymer complexes are a scalable alternative but can be structurally unstable, inefficient, and/or cytotoxic[7,8]. Reversible membrane disruption or poration can permit the efficient delivery of many different classes of molecules into a wide variety of cell types[9]. Electroporation is a well-established approach that uses brief pulses of intense electric field to create pores in biological membranes through dielectric breakdown. Electroporation can efficiently create small (i.e., 0.5–2 nm) hydrophilic pores (electropores)

across the plasma membrane, which are thought to remain thermodynamically stable even after the electric potential has been removed[10,11]. To deliver larger biomolecules, electropores can be temporarily enlarged by applying longer pulses, additional pulses, and/or using higher field intensities. However, permeability to large biomolecules tends to saturate due to increased conductivity of the membrane. Also, with these more intense E-fields come a range of negative secondary effects on cell integrity including DNA damage[12], lipid peroxidation[13], phenotypic changes and/or cell death[14,15]. From a manufacturing perspective, large-volume electroporation faces additional challenges associated with electrode corrosion, electrolysis, and heating, resulting in nonuniform electric field, pH changes, ionic contamination, and prolonged exposure of cells to unphysiologically low conductance solutions[9]. For example, recent Phase I clinical trials using high throughput electroporation systems (MaxCyte, Nucleofector LV, and/or ThermoFisher Xenon) for CRISPR-Cas9 gene editing of T cells have resulted in lower editing efficiencies than was reported by studies editing the same genetic locus in similar cells with laboratory scale electroporation[16–18]. Although some of the issues faced by large volume electroporation can be addressed with high throughput microfluidics[19,20], the intrinsic

[1]Center for Engineering in Medicine and Surgery, Department of Surgery, Massachusetts General Hospital, Harvard Medical School, Boston, MA 02114, USA. [2]Shriners Children's, Boston, MA 02114, USA. ✉e-mail: mehmet_toner@hms.harvard.edu

cytotoxicity associated with large biomolecule delivery by high-power electroporation remains unresolved.

Alternatively, the plasma membrane can be porated by applying one or more brief pulses of intense mechanical tensile stress, also called mechanoporation[21]. Compared with electroporation, mechanoporation is thought to result in fewer but larger pores, since mechanopores act as stress concentrations while electropores relax the nearby potential[22]. This would explain why mechanoporation can enable rapid delivery of large macromolecules and can be well tolerated by cells when precisely controlled[21,23–27]. However, highly controlled mechanoporation processes such as microinjection have proven difficult to scale. Techniques involving needles or some other surface physically contacting the membrane are effective for smaller numbers of cells but are susceptible to clogging and fouling[26,28–31]. To address this shortcoming, recent seminal studies into contact-free mechanoporation have demonstrated the feasibility of acoustic waves[32,33], fluid shear stress and inertia[25,34–37], or droplet encapsulation[38] to permit high throughput and continuous mechanoporation. While promising, most of these methods require cells to be dilute (greatly increasing reagent costs) while still achieving only modest transfection yield and throughputs that remain too low for clinical applications requiring billions of cells per dose (Supplementary Table 1).

Here we show a continuous, contact-free method for mechanoporation and biomolecule delivery which uses viscoelastic extensional flow to apply an approximately constant intensity and duration of membrane tension to all cells in a sample (Fig. 1). Cells are suspended in a viscoelastic solution with a biomolecule of interest. Within a microfluidic chip, cells are aligned by inertio-elastic focusing, preventing contact between cells and channel walls while also minimizing cell-to-cell variability in residence time and peak stress. Then, a geometric contraction is used to generate an extensional flow along the channel centerline. Cells within this extensional flow are deformed by fluid inertia and drag, resulting in mechanoporation and intracellular delivery of biomolecules.

## Results

### Viscoelastic mechanoporation is feasible for efficient intracellular delivery of large biomolecules

A microfluidic device with a single contraction-expansion channel was initially fabricated from polydimethylsiloxane (PDMS) to test whether a uniaxial extensional flow generated by a microfluidic contraction-expansion geometry could generate sufficient membrane tension for mechanoporation of Jurkat cells, an available T cell line. This device consisted of a single straight microchannel typically 100 μm long, 45 μm wide, and 50 μm tall, that connected two much larger chambers (1.5 mm wide, 2 mm long, and 50 μm tall) by tapered sections (Supplementary Fig. 1). The channel was therefore at least three times wider than the cells, which were between 9 μm and 14 μm (Supplementary

Fig. 2). Dextran labeled with fluorescein isothiocyanate (FITC−dextran) was initially used for optimization and characterization studies as an inert fluorescent dye of specified molecular size. For each experimental condition, about 500,000 cells were suspended in 50 μL of a delivery solution consisting of phosphate buffered saline (PBS), up to 3 mg/mL of 1.6 MDa hyaluronic acid (HA), and 0.2 mg/mL 2000 kDa FITC−Dextran, and then processed through the device by pressure-driven flow. Cells were immediately diluted tenfold in culture medium and incubated for one to two hours at room temperature before being counted, washed twice in PBS, stained with propidium iodide, and run on an imaging flow cytometer to assess cell viability and dextran delivery. Control samples were incubated in the delivery solution for a similar amount of time but not processed through the device.

We investigated the impact of HA concentration, operating pressure, channel geometry, and delivery solution composition on the uptake of FITC−dextran and cell recovery. Across all geometries and HA concentrations, the delivery efficiency (i.e., the percentage of viable cells which took up FITC−dextran) increased with increasing operating pressure, with efficiencies exceeding 90% for some configurations (Supplementary Figs. 3–5). In contrast, the delivery efficiency did not exceed 34% at any operating pressure if HA was excluded from the delivery solution. At the tested highest operating pressures the number of recovered cells decreased substantially compared to the unprocessed controls, which we attributed to lysis and/or irreversible mechanoporation (Supplementary Figs. 3d and 4c). The addition of calcium ion in the transfection solution was also evaluated, since prior work has established the role of $Ca^{2+}$ influx as an important signal for membrane repair[39–41]. Indeed, we found a slight but statistically significant increase in viability, from about 80% to about 90%, assessed 3 h after transfection, when calcium was included in the transfection solution (Supplementary Fig. 6).

### High throughput cell focusing enables high throughput and consistent cell deformation

We hypothesized that focusing all cells to the center of the channel would reduce process variability and cell loss. We developed and integrated an upstream inertio-elastic cell focusing module that consisted of two symmetrical outward-spiraling microchannels, each of which focused cells to an equilibrium position close to (but not touching) of the concave channel face (Fig. 2a–d). This layout also focuses cells halfway between the straight and parallel channel faces[42]. Thus, this module is designed to focus cells close to the centerline of the single downstream merged channel (Fig. 2b). A channel geometry with a large cross sectional aspect ratio and high curvature was designed to maximize focusing throughput, by both increasing the channel cross section and also delaying the breakdown of primary Dean vortices at higher flow rates[43]. These devices were fabricated from rigid epoxy bonded to glass to prevent channel inflation under high operating pressures (described in "Methods")[44]. We assessed the

*Narrowing microchannel*

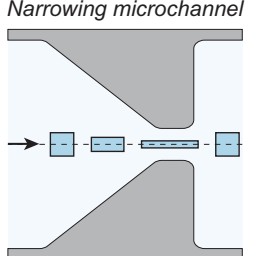

*Relative fluid motion*

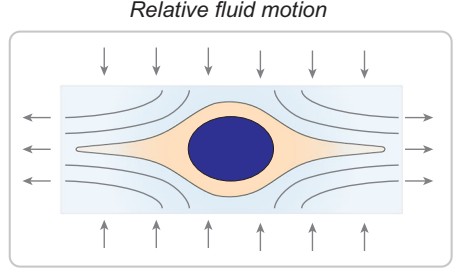

*Mechanoporation & Intracellular delivery*

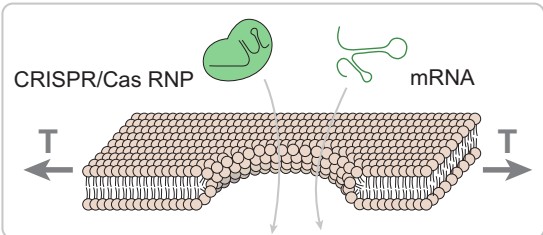

**Fig. 1 | Conceptual schematic for high throughput continuous-flow transfection by flow forces.** A fluid element traveling along the centerline of a narrowing channel becomes extended without rotation along the direction of motion. The plasma membrane of a cell within such a flow is stretched and reversibly permeabilized by inertial and drag forces from relative fluid motion, allowing intracellular delivery of biomolecules. The extensional viscosity of the solution is greatly increased by adding a biopolymer to make the solution viscoelastic, resulting in higher membrane tension at a given flow rate.

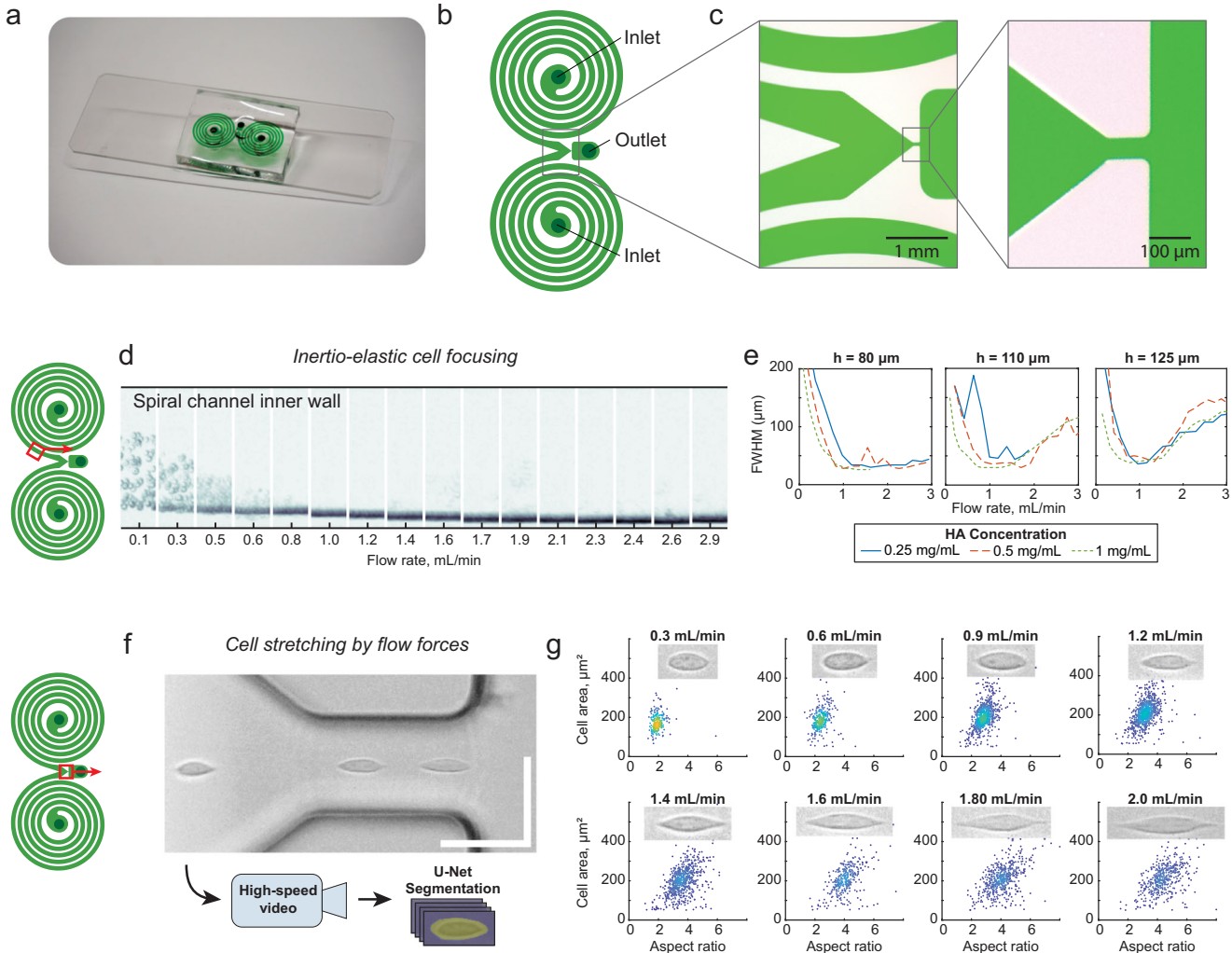

**Fig. 2 | High throughput and uniform cell stretching without surface contact.**
**a** Image of epoxy-glass microfluidic device with dye-filled channels. **b** Schematic of device layout and **c** device micrographs detailing the contraction-expansion section where the two spiral channels merge. **d** Composited time-lapse images of one of the spiral channels show cells being focused to a narrow band near the outer (concave) channel wall. Indicated flow rate is the total flow through both inlets. Composite images are 500 μm tall. **e** Full-width half-maximum (FWHM) of lateral cell position distributions across the width of the composite images, for a range of channel heights (h) and hyaluronic acid (HA) concentrations. Lower FWHM indicates tighter cell focusing. **f** High-speed video of cells in the narrowing channel were captured and segmented for quantitative shape analysis. Scalebar indicates 50 μm. **g** Scatterplots of measured cell area versus cell aspect ratio with selected representative inset images. The total numbers of imaged cells were 234, 312, 798, 830, 680, 324, 537, and 461 for flow rates of 0.3 mL/min, 0.6 mL/min, 0.9 mL/min, 1.2 mL/min, 1.4 mL/min, 1.6 mL/min, 1.8 mL/min, and 2.0 mL/min, respectively. Inset images are 16 μm tall. Source data are provided in the Source Data file.

focusing performance for Jurkat cells suspended in PBS with HA across a range of flow rates, HA concentrations, and channel heights. The same channel layout was used all tests (described in "Methods"). Cell focusing performance was visualized by creating composite time-lapse images from brightfield high speed microscopy videos and quantified by measuring the width of cell body distributions (described in "Methods"). All tested combinations of channel geometry and HA concentration resulted in successful focusing at flow rates close to 1 mL/min, with higher HA concentrations resulting in more efficient focusing at lower flow rates (Fig. 2e and Supplementary Fig. 7). A channel height of 80 μm was selected for further study as it exhibited excellent focusing performance across the largest range of flow rates and polymer concentrations, particularly at higher flow rates where de-focusing was observed in the channels with 100 μm and 120 μm heights.

The deformation of cells within the contracting microchannel were imaged by high-speed video microscopy and quantified using a convolutional neural network (Fig. 2f, g, described in "Methods"). As expected, cells were elongated along the flow direction without

contact with channel walls, and the cell aspect ratio increased with increasing flow rate. Qualitatively, at lower strain rates cells remained rounded, while at higher strain rates cells were transiently pulled into an elongated spindle-like morphology, consistent with previous studies of transient cell deformation in pure extensional flows[25,35,38]. Above 2 mL/min, images were degraded by motion blur due to very high flow speeds (about 10 m/s) and a minimum exposure time of about 0.3 μs.

## Flow visualization and computational studies reveal vortices
The flow kinematics in the contracting channel were visualized by flowing 2 μm polystyrene beads dispersed in PBS with 0.5 mg/mL HA through the devices. Particle pathlines revealed symmetric flow separation from the contracting channel walls beginning at about 0.05 mL/min, followed by symmetric vortex growth up to about 200 μL/min and asymmetric oscillating vortices at higher flow rates (Fig. 3a). The upstream length of the vortices increased with increasing flow rate (Fig. 3b). Qualitatively, the frequency of oscillations also increased with increasing flow rate.

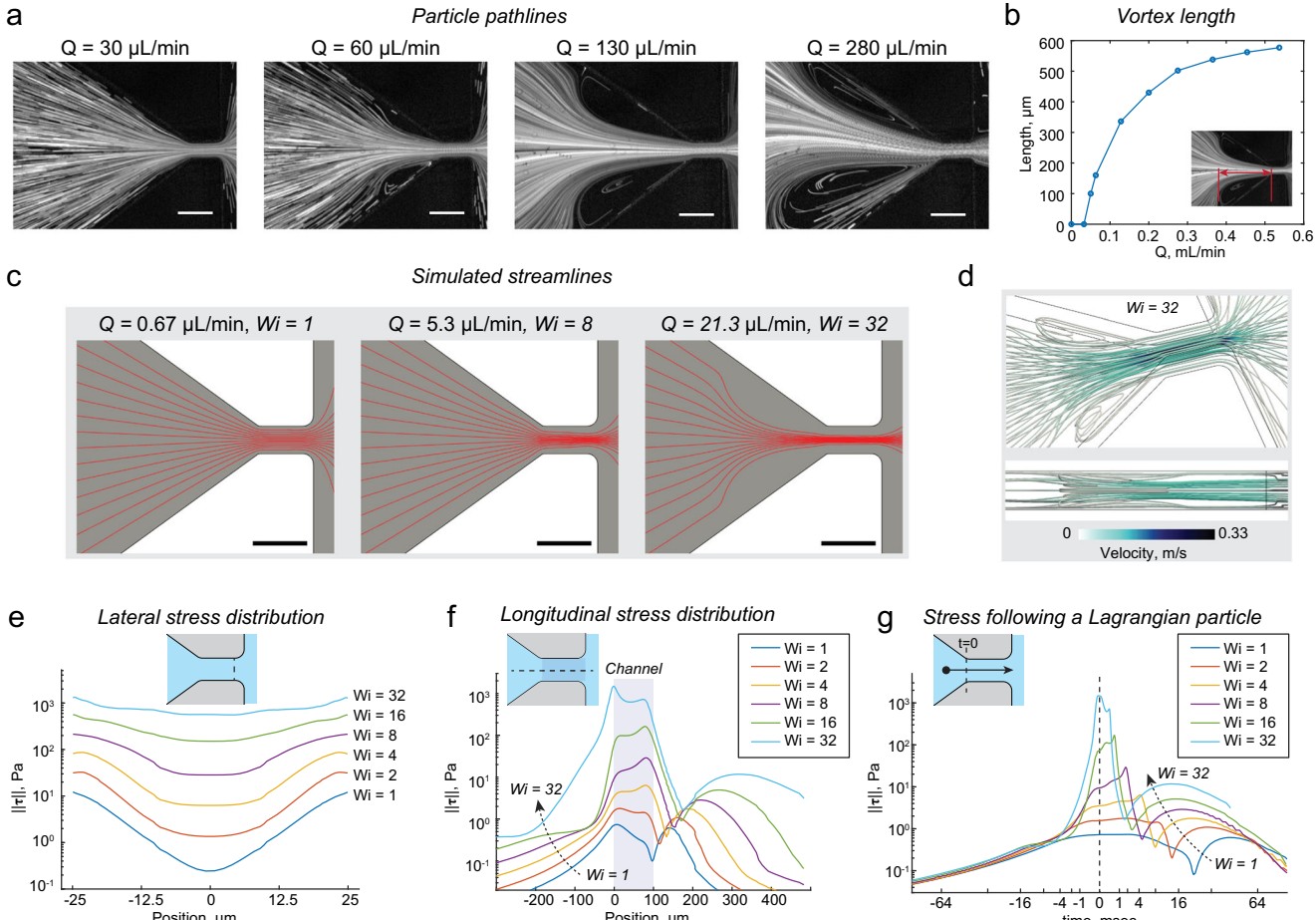

**Fig. 3 | Experimental and computational characterization of the flow.**
**a** Experimentally measured particle pathlines show flow separation and vortex growth beginning at about 50 μL/min. **b** Vortex length as a function of flow rate, with vortex length measurement annotated in the inset image. **c** Simulated streamlines showing deviations from a Newtonian flow profile at a Weissenberg number ($Wi$) of 8, and upstream flow separation at $Wi = 32$. **d** Simulated streamtubes for $Wi = 32$ show 3D flow pattern. **e**, **f** From simulations, line cuts of the spectral norm of the deviatoric stress ($\|\boldsymbol{\tau}\|$, described in text) for a range of Weissenberg numbers. **g** Simulated histories of $\|\boldsymbol{\tau}\|$ for an idealized Lagrangian particle which enters the constriction at time $t = 0$ msec. Scalebars in **a** and **c** indicate 100 μm. Source data are provided in the Source Data file.

We also performed 3D computational fluid dynamics simulations of PBS-HA solutions in the contraction as a FENE-CR (finite extendable non-linear elastic−Chilcott and Rallison) fluid using RheoTool[45]. Details about the simulation configuration are provided in the "Methods". The range of flow rates accessible to computational study were still limited to less than 1% of those required for mechanoporation, due to fundamental numerical challenges associated with fast viscoelastic flows (discussed below). Also, the simulated geometry was not identical to the experimentally optimized one (50 μm tall in simulations, versus 80 μm tall in experiments). Nevertheless, the simulation results recapitulated key features of the flow kinematics observed in experiments and elucidated the likely distribution of fluid internal stresses. The Weissenberg number $Wi$ is a dimensionless number which compares the magnitude of elastic and to viscous stresses. Simulations were conducted for flow rates corresponding to channel Weissenberg numbers $Wi$ of 1, 2, 4, 8, 16, and 32. Dimensionless numbers are defined and discussed in Supplementary Note 1.

In the simulations, the onset of flow separation was observed at about 20 μL/min, but was preceded by deviations from the Newtonian laminar flow profile within the contraction starting at about 4 μL/min. We noted that the onset of deviations corresponded precisely with regions of the flow where the local elastic Mach number exceeded one (Supplementary Note 2 and Supplementary Fig. 8).

To visualize the spatial distribution of internal stresses we plotted the spectral norm $\|\boldsymbol{\tau}\|$ of the deviatoric stress tensor $\boldsymbol{\tau}$ (i.e., the magnitude of the largest eigenvalue of the $\boldsymbol{\tau}$). For a hypothetical small particle, $\|\boldsymbol{\tau}\|$ may be considered analogous to the first principal stress. $\|\boldsymbol{\tau}\|$ is relevant here because it is the extensional component of any given flow field that is primarily responsible for cell stretching and mechanoporation[46]. For each simulated case, we have plotted the distribution of $\|\boldsymbol{\tau}\|$ across the width of the channel (Fig. 3e) and along the channel centerline (Fig. 3f). All data are sliced from the channel center plane, i.e., halfway between the parallel upper and bottom faces of the channel. We have also plotted the stress history of $\|\boldsymbol{\tau}\|$ for a Lagrangian particle following the center streamline (Fig. 3g). At low flow rates ($Wi = 1$), drag from the walls was dominant, and stresses along the channel centerline were much less than those at the walls. At higher flow rates, the contributions to stress from the upstream contraction grew faster than contributions from wall drag, resulting in a more uniform lateral stress distribution (Fig. 3e). Also, while at $Wi = 1$ the stresses associated with flow contraction and expansion were comparable, at high flow rates the stresses associated with flow acceleration exceeded that of deceleration by over two orders of magnitude.

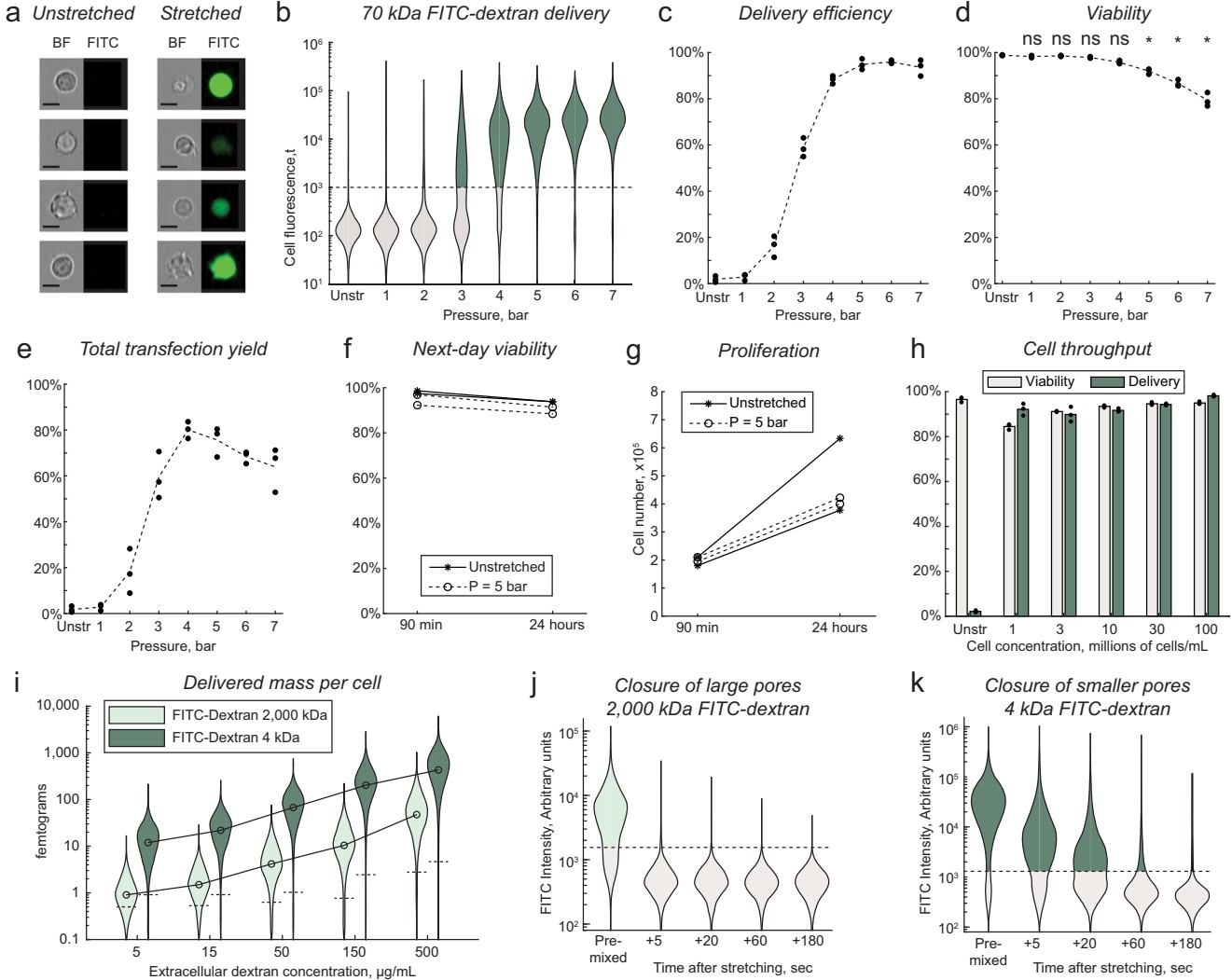

**Fig. 4 | Optimization and characterization of mechanoporation for intracellular delivery. a** Representative brightfield (BF) and fluorescence (FITC) images of Jurkat cells after viscoelastic mechanoporation and delivery of 70 kDa FITC–dextran at 5 bar. Representative images were selected from 7302 images of cells in this sample. Images are padded to uniform size. Scalebars indicate 10 μm. **b** Distributions of intracellular FITC–dextran fluorescence of viable cells after processing through the chip at a range of pressures. **c** Delivery efficiency indicates the percentage of viable cells brighter than the arbitrary cutoff indicated by the dotted line in (**b**). Unstr indicates unstretched, i.e., cells were incubated with the delivery solution containing FITC–dextran but not processed through the chip ($n = 3$ replicates per condition). **d** Cell viability assessed by propidium iodide exclusion about 90 minutes after processing ($n = 3$ replicates per condition). ns indicates no significant difference from unstretched control, * indicates significantly different from unstretched control by $p<0.05$ criterion using one-way ANOVA and Tukey honest significance difference correction, with $p = 1.5 \times 10^{-4}$ for

5 bar, $p = 1.4 \times 10^{-7}$ for 6 bar, and $p = 6.0 \times 10^{-8}$ for 7 bar. **e** Total transfection yield is the number of viable and dextran positive cells recovered, as compared to the average number of viable cells recovered in the unstretched control samples ($n = 3$, relative number of recovered cells provided in Supplementary Fig. 10). **f** Viability and **g** total cell number following 90 minutes or 24 h of culture after processing ($n = 2$ replicates per condition). **h** Viability and dextran delivery for samples where the cell concentration was increased ($n = 3$ replicates per condition). **i** Distributions in the total mass of delivered dextran per cell, for large and small dextran species at a range of extracellular concentrations. **j, k** Distributions in the amount of delivered dextran per cell when the dextran is added following a delay after cell stretching, rather than premixed with the solution before stretching as usual. Dotted lines in (**b, i–k**) indicate arbitrary thresholds for positive delivery, i.e., 98th percentile of unstretched controls. Dotted lines in **c–e** and bar plots in **h** indicate replicate mean value. Source data are provided in the Source Data file.

## Optimization of viscoelastic mechanoporation for high throughput and uniform intracellular delivery

We next optimized the delivery efficiency, viability, cell recovery, and throughput performance of the device with integrated focusing (Fig. 4a–h). Cells were suspended in PBS with 0.5 mg/mL HA and 0.2 mg/mL FITC–Dextran 70 kDa. The device was operated at a range of pressures from 1 bar to 7 bar, corresponding to flow rates from 0.5 mL/min to 4 mL/min (Supplementary Figs. 9 and 10). As observed previously, the delivery efficiency increased with increasing flow rate to a maximum of about 95% of cells at 5 bar (Fig. 4d). Viability assessed the same day did not significantly decrease at pressures below 5 bar

but decreased slightly but significantly thereafter, with viability decreasing further with increasing operating pressure (Fig. 4c). Unlike with the contraction-only device, the number of recovered cells did not significantly decrease at any tested flow rate (Supplementary Fig. 11), suggesting cell lysis had been effectively prevented by cell focusing. An overall yield was defined as the number of viable and dextran positive cells recovered as compared to the number of viable cells recovered in unstretched samples (Fig. 4e). An operating pressure of 5 bar was selected for further study as the minimum pressure required to achieve complete transfection (i.e., about 95% of cells). At this pressure, the flow rate through the chip was about 2.7 mL/min. No

correlation was observed between cell size and FITC−dextran delivery or viability. After 24 h in culture, cell viability was still above 90% for all samples and the cell number was about twofold higher, suggesting successful cell recovery and proliferation (Fig. 4f, g and Supplementary Fig. 12).

We next evaluated whether increasing cell concentration in the delivery solution could be used to increase throughput. The delivery and viability did not decrease with increasing cell concentrations even at the highest tested concentration of 100 million cells per mL, supporting the feasibility of high throughput intracellular delivery at over 250 million cells per minute (Fig. 4h). To assess the feasibility of larger volume transfections, we also ran larger 500 μL volumes each containing 25 million cells. We saw the transfection efficiency and viability remained greater than 90% when measured later the same day (Supplementary Fig. 13).

We next performed quantitative flow cytometry to measure the mass of delivered FITC−dextran for both small and large FITC−dextran species as a function of dextran concentration in the delivery solution (Fig. 4i, calibration data Supplementary Fig. 14). Quantitative flow cytometry is more accurate for quantifying delivery than fold-change cell fluorescence, which is affected by background sources that vary between imaging systems, cell types, and sample types including autofluorescence and instrument background. Across all extracellular concentrations, the average amount of delivered FITC−dextran per cell was about 10-fold higher for the smaller species (4 kDa) than for the larger one (2000 kDa). If the cell was roughly approximated as a 12 μm sphere (based on the average cell diameter, Supplementary Fig. 2), the average intracellular concentration of 4 kDa dextran fell within a factor of two of the extracellular concentration for all tests (Supplementary Fig. 15). To assess how long the cell membrane remained permeable after mechanoporation, cells were processed through the chip without FITC−dextran in the solution, and then either small (4 kDa) or large (2000 kDa) FITC−dextran was spiked into the solution after a delay of up to three minutes (Fig. 4j, k). Some uptake of 4 kDa FITC−dextran was observed 20 s after processing, albeit greatly reduced, while uptake of 2000 kDa FITC−dextran was undetectable even when it was added just five seconds after processing.

To verify that intracellular delivery was not restricted to endosomes, Jurkat cells were labeled with a covalent membrane dye immediately prior to delivering 70 kDa FITC−dextran by viscoelastic mechanoporation. Cells were fixed immediately after processing through the chip and imaged using a confocal fluorescence microscope (Supplementary Fig. 16). FITC−dextran was observed throughout the cytosol of only transfected cells and was not restricted to newly formed endosomes. We also compared this at the population level by measuring the intracellular distribution of 70 kDa FITC−dextran after delivery by viscoelastic mechanoporation against an endocytosis positive control, where cells were incubated with 1 mg/mL of the dye for 90 min under culture conditions. ImageStream imaging flow cytometry revealed the dye was distributed more evenly throughout the cell following mechanoporation versus endocytosis, which was clearly visible in the raw images as well as by two different ImageStream metrics of dye distribution (Supplementary Fig. 17). To determine whether viscoelastic mechanoporation resulted in any longer-term alterations to cell shape or membrane morphology, we compared three different ImageStream metrics of plasma membrane morphology and fluorescence intensity distribution about 15 min after processing through the chip to those of unstretched cells (Supplementary Fig. 18). Altogether, no substantial differences were observed between stretched and unstretched cells. To verify that intracellular delivery by viscoelastic mechanoporation was due to viscoelastic fluid properties rather than a biochemical effect specific to HA, viscoelastic solutions were prepared using 2000 kDa polyethylene oxide (PEO) instead of HA at concentrations of 0.2 mg/mL and 1 mg/mL in PBS. Consistent with previous results, delivery of 70 kDa FITC−dextran increased with increasing operating pressures, with efficient (>85%) delivery observed at some operating pressure for both tested concentrations of PEO (Supplementary Fig. 19).

## Viscoelastic mechanoporation delivers protein, mRNA, and ribonucleoprotein complexes to Jurkat cells

We next evaluated whether viscoelastic mechanoporation could deliver broader categories of biomolecules. We found that FITC-tagged albumin could be delivered with an efficiency of about 90% using the optimized protocol (Supplementary Fig. 20). We next tested delivery of mRNA encoding enhanced green fluorescent protein (eGFP) for a range of extracellular mRNA concentrations. 30 μg/mL to 100 μg/mL is a typical concentration range for mRNA delivery by electroporation or mechanoporation[19,20,47,48]. Viability and eGFP expression were evaluated 24 hours after processing. As expected, eGFP expression increased with increasing extracellular concentrations of mRNA (Fig. 5a). Following delivery of 100 μg/mL mRNA, 89% of viable cells were eGFP positive compared to 2% of cells that were incubated with the 100 μg/mL of mRNA but not processed through the chip. Viability decreased with increasing mRNA concentration in the delivery solution to a minimum of 74% for 100 μg/mL, perhaps due to changes in the solution rheology due to the addition of the mRNA which is itself a polymer (Supplementary Fig. 21).

RNP delivery for gene editing was evaluated in Jurkat cells using a synthetic crRNA sequence previously optimized to knock out the T cell receptor (TCR, Supplementary Table 2)[18]. RNPs were complexed in vitro at a ratio of 1:2:2 Cas9:crRNA:tracrRNA, then added to the delivery solution at a final Cas9 concentration between 0.1 μM and 1 μM. RNP were delivered using the optimal protocol and cultured for two days. TCR expression was assayed by immunofluorescence staining and flow cytometry. As expected, the percentage of cells expressing detectable levels of TCR decreased with increasing RNP concentrations, from about 85% of untreated cells to less than 40% of cells treated with 0.3 μM or 1 μM RNP, indicating an average knockout efficiency of about 53% (Fig. 5b). The viability of cells transfected with RNP decreased to between 66% and 73% after 2 days, although without a dose-dependent trend, which suggested some degree of cell damage associated with RNP delivery by mechanoporation (Supplementary Fig. 22).

## Viscoelastic mechanoporation of HEK cells and primary T cells

We next evaluated whether viscoelastic mechanoporation could be used to deliver molecules to other cell types, such as adherent cells or primary cells. We first optimized a protocol for delivery of 70 kDa FITC−dextran to HEK293T cells, an available adherent cell line. Cells were initially processed in 0.5 mg/mL HA in PBS at a range of operating pressures. Efficient delivery was observed at the highest operating pressures (>90%), although viability after about 90 min was below 50% (Supplementary Fig. 23). We hypothesized that viability may be improved by using a cytoplasmic buffer in place of PBS, that would approximate the ionic composition of the cytosol[49]. We noted that compared to Jurkat cells, HEK293T cells are also generally larger and may have different cytoskeletal mechanical properties. We were also interested to evaluate whether extracellular calcium would be helpful or harmful to cell recovery, as $Ca^{2+}$ influx is a required signal for membrane repair but can also trigger apoptosis. Overall, the cytoplasmic buffer with supplemented calcium was found to be best, resulting in viability after 24 h similar to that of the untreated controls, and delivery efficiency of about 88% (Fig. 5c). Finally, we screened a range of HA concentrations and flow rates for delivering 70 kDa FITC−dextran to primary activated T cells (Supplementary Fig. 24). Ultimately, the highest tested HA concentration of 2 mg/mL was best, with an average delivery efficiency of 84% and average viability of 85% with an operating pressure of 8 bar (Fig. 5d).

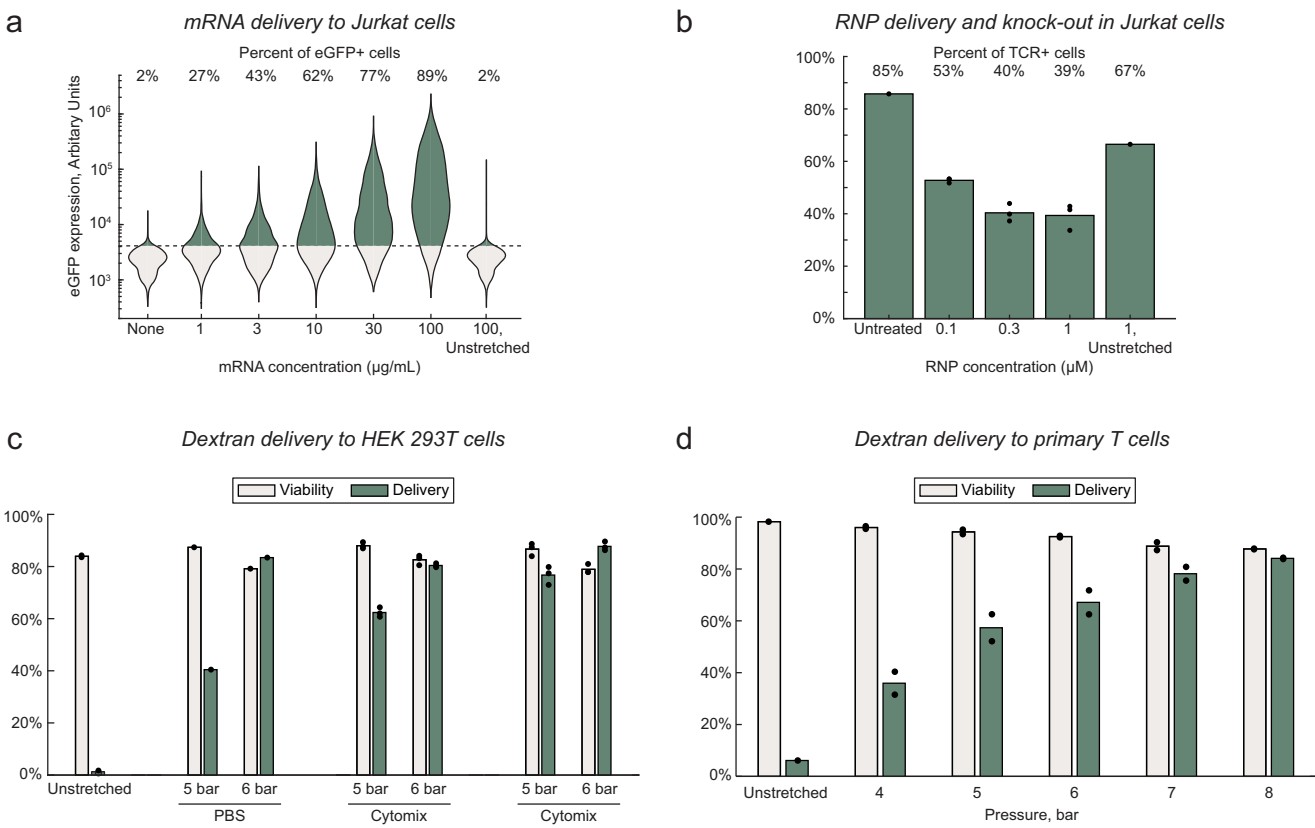

**Fig. 5 | Generalization to different cargo molecules and cell types. a** eGFP expression 24 h after mRNA delivery to Jurkat cells for a range of extracellular mRNA concentrations. **b** T cell receptor (TCR) expression in Jurkat cells 2 days after delivery of Cas9 RNP complexes with guide RNA to knock out TCR expression, for a range of extracellular RNP concentrations (experimental conditions, $n = 3$; controls, $n = 1$ replicates per condition). **c** Viability and delivery of 70 kDa FITC−dextran to

HEK293T cells, 24 h after processing, for three delivery buffer compositions and 2 different chip pressures (Unstretched, $n = 2$; PBS, $n = 1$; cytomix buffers, $n = 3$ replicates per condition). **d** Viability and delivery of 70 kDa FITC−dextran to primary activated T cells, about 90 min after processing, for increasing chip pressures (test conditions, $n = 2$; unstretched control, $n = 1$ replicate per condition). Source data are provided in the Source Data file.

## Discussion

In this work, we developed a high throughput method of applying consistent mechanoporation conditions to cells without surface contact. Cells were stretched and permeabilized using a viscoelastic extensional flow generated by a contracting microchannel. Contact between cells and walls was actively prevented by a high throughput cell focusing module, enabled using rigid prototyping materials. Efficient intracellular delivery was accomplished for suspension (Jurkat) and adherent (HEK293T) cell lines as well as primary T cells, supporting the feasibility of this method as a general approach for high throughput intracellular delivery. In each case the same device was used, and delivery and cell viability were optimized by adjusting the flow rate and HA concentration. Although high speed microscopy images suggested a correlation between the size and aspect ratio of stretched cells, no correlation between cell size and mechanoporation was observed in the flow cytometry results. Buffer composition could be optimized to improve cell viability and recovery without restrictions in conductivity, as in electroporation. Efficient mRNA expression was observed at extracellular mRNA concentrations similar to those typically used for electroporation. For applications invovling nucleic acids and/or proteins, process costs were dominated by costs of the transfection reagents (Supplementary Table 3). Altogether, viscoelastic mechanoporation seems to be a feasible approach for efficient intracellular delivery at a throughput of over 250 million cells per minute.

This approach should be compared to other recent developments in mechanoporation technologies (compared quantitatively in Supplementary Table 1). Important early studies characterized

the transient membrane disruption caused by mechanical cell squeezing[50]. More recently, cells have been efficiently porated in microfluidic systems that included cross-slot flows[36], T junctions[25,35], vortices[37,51,52], and nebulizers[32]. Cells have also been observed to deform in highly viscous and/or shear-thinning solutions, and this was recently explored for improving delivery during microfluidic cell squeezing[29,53]. These seminal studies established the feasibility of mechanoporation for intracellular delivery. The important differences between viscoelastic mechanoporation and these other methods are the process uniformity, throughput, and delivery performance. Prior flow-based mechanoporation strategies also did not maintain a consistent stress history for all cells. This was because the flow conditions were either inhomogeneous (i.e., spatially varying, with cells in different positions experiencing different loads) or singular (i.e., cells were hydrodynamically trapped for uncontrolled amounts of time at a stagnation point or vortex). Relatedly, none of these methods were compatible with high cell concentrations (e.g., $10^8$ cells/mL) due to issues with clogging or the fact that only one cell should occupy the stagnation point at a time. In this work, clogging and uniformity issues were avoided by focusing cells to the center of a channel that was several times wider than the cells. The favorable performance and high throughput of viscoelastic mechanoporation were enabled by leveraging intrinsic viscoelastic properties of dilute polymer solutions to achieve sufficient membrane tensions in a continuous-flow system. Recently, Kwon and Chung showed efficient mechanoporation with a shear-thinning solution for the first time[29]. While impressively high delivery was demonstrated, this

approach relied on a different non-Newtonian flow phenomenon (shear thinning rather than viscoelasticity, discussed further in Supplementary Note 3: A large contraction ratio is required for strain hardening) and cell throughput and cell viability were about 500× and 2× lower, respectively, than for viscoelastic mechanoporation.

Several recent studies have identified differences between mechanoporation and electroporation at both the cell recovery and gene expression levels. In particular, Hu and colleagues recently used RNA sequencing to show flow-based mechanoporation of primary T cells resulted in not only improved cell viability but also reduced genetic dysregulation as compared to laboratory-scale electroporation systems[54]. Likewise, Jarrell and colleagues showed flow-based mechanoporation resulted in better viability, proliferation, and cyto-kine secretion in T cells when compared with electroporation[37]. Alto-gether, these studies suggest mechanoporation can achieve similar or better delivery with less cellular damage than electroporation in T cells, although more work will be required to elucidate the mechanisms involved. On the other hand, mechanoporation strategies including this work are limited in comparison to electroporation in their capacity to selectively target intracellular compartments and organelles, and the inability to electrophoretically drive charged bio-molecules such as nucleic acids into the cytoplasm or into contact with the plasma membrane[9]. Emerging hybrid electro-mechanical strate-gies are a promising avenue of investigation for maintaining the advantages of electroporation for delivering large nucleic acids, for example, while reducing cell damage[20,55].

Results were most consistent with the hypothesis that the mechanism of delivery was biomolecule diffusion into the cytosolic compartment through short-lived mechanopores. While membrane stretching can promote endocytosis, the effect is moderate (i.e., doubling the baseline rate of endocytosis)[56]. Here we showed over hundredfold greater delivery of relatively large biomolecules versus unstretched (endocytosis-only) controls, and confocal imaging con-firmed delivery throughout the cell.

The computational studies were limited to relatively low flow rates due to the so-called high Weissenberg number problem, a numerical stability issue that makes simulations of fast viscoelastic flows computationally expensive[57]. Although inertio-elastic drag reduction in planar or Taylor–Couette geometries have been simulated for high Reynolds numbers[58–60], virtually all prior com-putational studies of viscoelastic contracting flows have been 2D and/or neglected fluid inertia[61–63]. Simulations in this work clarified that the timescale of peak membrane tension was likely comparable to the residence time within the narrow channel (i.e., of order 10 μs at 2.7 mL/min), and that the peak stress was likely much greater than the minimum steady-state stress required for red cell lysis (about 150 Pa or 1500 dyne/cm$^2$)[64] since a peak stress of 1000 Pa was esti-mated at just 1% the flow rate required for mechanoporation. Experimental studies of lipid membranes under dynamic loads have previously shown that membranes can resist anomalously high tensions on shorter timescales[22]. At higher flow rates an unsteady flow condition was observed but did not prevent effective mechanoporation, perhaps because peak stresses were expected to be limited to within the narrowest portion of the contraction where laminar flow is restored. Nevertheless, avoiding this potential source of variability may further improve the consistency of the mechanoporation process.

The possibility of delivering the polymer into the cell was not investigated. Regarding regulatory considerations, hyaluronic acid is considered safe to use for several therapeutic indications including visco-supplementation and as dermal filler but its regulatory status in the context of cell and gene therapy production has not been established[65]. Here we showed that the method requires a viscoelastic solution but is not dependent on a specific chemical structure. This will permit material selection and optimization for different cell types and clinical applications as required.

Altogether, this work demonstrated that viscoelastic mechan-oporation is feasible for high throughput delivery of biomolecules into mammalian cells ex vivo. Further optimization of the device geometry will mitigate inertio-elastic instabilities, while selection of different viscoelastic polymers will allow cell- or process-specific optimization for diverse biomedical applications. We expect these studies to inspire and enable further development of scalable intracellular delivery strategies for cell and gene therapy manufacturing.

## Methods

### Ethical statement
Our research complies with all relevant ethical regulations. This work did not include human subjects research. Experimental protocols involving the use of anonymized human derived samples from healthy donors have been approved by the Mass General Brigham Institutional Biosafety Committee, record number 2012B000060.

### Microfluidic device layout, fabrication and fixturing
The convex and concave walls of the spiral microchannels were parametrically defined in cylindrical coordinates by $R(t) = (0.75 \, \text{mm})t$, $\Theta(t) = 2\pi t$ and $R(t) = (0.75 \, \text{mm}) \, t + 0.5 \, \text{mm}$, $\Theta(t) = 2\pi t$, respectively, for $2 < t < 7$, resulting in a fixed width of 500 μm that widened gradually to 0.75 mm at the confluence. Devices were microfabricated from molded epoxy bonded to glass. Master molds were fabricated by photolithographic patterning of spin coated SU-8 films on silicon wafers. Then, poly(dimethyl) siloxane (PDMS, DOW Sylgard™ 184) replica molds were prepared by pouring uncured PDMS over the master mold, curing overnight at 80 °C, releasing, and cutting to size. PDMS replicas were placed face up in a plastic petri dish and passivated by Trichloro(1H,1H,2H,2H-perfluorooctyl)silane (Sigma Aldrich #448931) vapor deposition surface treatment for 15–30 min in a vacuum chamber. PDMS was poured onto the face-up replicas and cured overnight to form a secondary mold. The secondary mold was peeled from the petri dish, inverted, and PDMS replicas removed. The secondary mold was prepared by punching 1.5 mm holes in the locations of the inlets and outlets, and 5 mm segments of 0.07" polytetrafluoroethylene (PFTE) rod (McMaster-Carr, #84935K52) were inserted into the holes. On the other end of each PFTE rod, a 1.5 cm segment of 1/16th in. ID polyurethane tubing (McMaster-Carr #5648K67) was inserted. Two-part epoxy (Smooth-On, Inc., Epox-ACast™ 690) was mixed following the manufacturer instructions and poured into the mold. The epoxy part was removed from the mold after 20–24 h and bonded to plasma-treated glass slides by light pressure. Devices were allowed to cure for at least 2 days before use. 1/16th in. barbed fittings with Luer–Lok adapters were attached to the inlet tubing. Custom inserts were 3D printed from Tough 1500 resin (Formlabs #RS-F2-TO15-01) at 25 μm resolution (Form 3B, Formlabs) and attached to the fittings to create sample reservoirs. Finally, 1/16th in. outer diameter, 0.03 inch inner diameter PEEK tubing (IDEX #P-712) was fitted into the outlet tubing to reduce the outlet tubing volume.

### Preparation of viscoelastic solutions
Stock solutions of 4 mg/mL HA were prepared by dissolving 1.6 MDa sodium hyaluronate (HA15, Lifecore Biomedical) in PBS overnight with gentle rocking at room temperature. HA solutions were stored away from light at 4 °C and used within 2 weeks. Stock solutions of 5 mg/mL polyethylene oxide, 2 MDa (Sigma-Aldrich #372803) were prepared and stored similarly.

### Cell segmentation for automated cell deformation analysis
High-speed videos were acquired of cells passing through the con-tracting microchannel using brightfield phase contrast microscopy with a 40× objective. Videos were subsampled as necessary to ensure

no individual cell appeared more than once. Altogether about 48,000 frames were acquired across eight different tested flow rates. Training data was generated by randomly selecting a subset of 3000 frames and manually drawing individual cell outlines in ImageJ. The standard U-Net classifier was used, with outputs weighted by the prevalence of cell-pixels versus background-pixels in the training data[66]. Left–right and top–bottom reflections of the training dataset were used for data augmentation. Model training took about 18 h, and automated image segmentation took about 1 s per image (i.e., about 30 h for the entire dataset).

### Delivery experiments

Jurkat clone E6-1 (ATCC, TIB-152) and HEK293T (ATCC 293T/17) cell lines were used in studies as indicated. Except where stated otherwise, cells were resuspended at concentrations between 3 million cells/mL and 10 million cells/mL in a delivery solution consisting of PBS, 0.2 mg/mL 70 kDa FITC–dextran, 0.5 mg/mL 1.5 MDa HA, and 0.2 mM $CaCl_2$. The cell suspension was strained with a 40 μm nylon mesh immediately prior to processing through the chip. The cytoplasmic buffer used for HEK293T cells consisted of 100 mM $KH_2PO_4$, 15 mM $NaHCO_3$, 12 mM $MgCl_2 \times 6H_2O$, 8 mM ATP, and 2 mM glucose, to which with either 2 mM $CaCl_2$ or 2 mM Ethylenediaminetetraacetic acid (EDTA) were added. For each experimental condition, 100 μL of the mixture was pipetted into each of the two 3D printed reservoirs. The chip was primed with the sample solution by attaching an airtight pneumatic manifold to the tops of both reservoirs and using gentle pressure via a hand-operated syringe filled with air. To run the sample, the reservoirs were pressurized by opening a pneumatic valve to a regulated pressure source. In this way, 200 μL samples were typically processed within about 5 s. The sample exiting the outlet tubing was typically collected in an empty 1.5 mL microcentrifuge tube and a fixed volume (e.g., 150 μL) pipetted immediately into room temperature culture medium (for Jurkat cells and T cells, RPMI 1640 supplemented with 10% fetal bovine serum and 100 u/mL pen-strep; for HEK293T cells, DMEM supplemented similarly). For unstretched controls, a similar volume of cells (i.e., 150 μL) was diluted into culture medium without first processing through the chip. Cells were maintained at room temperature for up to 1 h before being washed twice in fresh culture medium. The delivery efficiency was defined as the percentage of viable (i.e., PI-negative) cells with a higher FITC signal than the substantial majority (e.g., 97–98%, manually gated) of viable cells in an unstretched control sample, for which cells were incubated in the delivery solution containing the dye but not processed through the chip. This permitted endocytotic uptake or cell surface binding to be distinguished from mechanoporation and intracellular delivery. All flow cytometry analysis was performed using Amnis IDEAS software version 6.2. An example of the flow cytometry gating scheme is provided (Supplementary Fig. 25).

### mRNA transfection, Cas9 RNP transfection, and TCR expression immunocytochemistry

For mRNA transfection, we used an mRNA encoding enhanced green fluorescent protein (eGFP) with ARCA cap modifications (Apexbio Technologies, Fisher Scientific #50-199-8310). For Cas9 RNP complex transfection, 24 μL of 60 μM crRNA (Supplementary Table 2) was hybridized with 12 μL of 120 μM atto550-tagged Atr-R™ tracrRNA (Integrated DNA technologies) in Tris-buffered saline by warming to 80 °C for 5 min, then cooling to 4 °C for 15 min. The hybridized RNAs were then mixed with 25 μL of 30 μM (5 mg/mL) recombinant SpCas9 protein (Sigma-Aldrich #Cas9PROT) for 30 min at 4 °C, resulting in a molar ratio of 2:1:1 Cas9:crRNA:tracrRNA. T cell receptor (TCR) expression was quantified by FITC anti-human TCR α/β antibody (BioLegend #306706, clone IP26) and imaging flow cytometry. For each condition, about 100,000 cells were incubated in 100 μL of PBS with 0.1% bovine serum albumin (BSA, Sigma Aldrich #A9418) and 5 μL of antibody (i.e., 1:20 dilution) on ice for 30 min, then washed twice in chilled PBS with BSA. Viability, eGFP expression, and TCR immunocytochemistry were assessed by flow cytometry following the same gating strategy as prior studies with FITC–dextran, as described above and shown in Supplementary Fig. 25.

### Isolation and culture of primary immune cells

Whole blood in EDTA tubes collected from healthy donors was purchased from Research Blood Components, LLC (Watertown, MA). Human peripheral blood mononuclear cells were isolated the same day by density centrifugation with Ficoll–Paque, and CD3 T cells were isolated from the PBMC fraction by incubation with anti-CD3/anti-CD28 magnetic beads following manufacturer's instructions (Dynabeads, Thermo Fisher Scientific #11161D). Activated T cells were subcultured on day +3 and when cell density exceeded 1e6 cells/mL. Delivery experiments were uniformly conducted on day +7 after isolation.

### Flow visualization studies

Polystyrene tracer particles (2 μm, Stokes number <0.001) were dispersed at 0.05% w/v in 0.5 mg/mL HA in PBS and imaged with a high-speed camera at 64,000 frames per second. Particle pathlines were generated by projecting the pixelwise standard deviation through several hundred sequential frames in MATLAB.

### Computational fluid dynamics simulations

We used RheoTool, a package extending the capabilities OpenFOAM with stabilized numerical solvers and high resolution schemes optimized for viscoelastic flows[45]. OpenFOAM version 9 was used. A high resolution fully swept mesh was generated, consisting of about 2,390,400 hexahedral elements, that followed the experimental geometry except that the channel height was set to 50 μm rather than 80 μm (Supplementary Fig. 26). The viscoelastic rheological parameters $\lambda = 0.0125$ sec (relaxation time) and $\eta_0 = 5$ mPa · sec (zero-strain viscosity) were selected based on published rheological measurements of aqueous solutions of 1.5 MDa hyaluronic acid with a concentration of about 1 mg/mL[67–69]. Together with a characteristic channel shear length scale of $d = 25$ μm and fluid density of $\rho = 1000$ kg/m$^3$, these provided a characteristic Elasticity number of exactly 100 (Supplementary Note 1). The FENE-CR constitutive model was used, with the solvent and polymer contributions to the viscosity selected as $\eta_s = 1$ mPa · sec and $\eta_p = 4$ mPa · sec, respectively, and the chain length factor was set to 1000, which is considered large for a FENE-type model (i.e., the simulated chains are long) but was motivated by the biophysical chain length of 1.6 MDa hyaluronic acid being roughly 4000 monomers per molecule[70].

### Quantitative flow cytometry

The Amnis ImageStream MkII performs multispectral imaging using multiple laser excitation and a dichroic filter stack to perform spectral decomposition onto a single CCD sensor. For green-emitting fluorochromes, emission is collected from 505 nm to 560 nm with a band peak at 533 nm. For FITC excitation, a 488 nm laser is used. The Ultra Rainbow Calibration bead kit (URCP-38-2k, Spherotech, Lake Forest, IL) consists of 3.8 μm polystyrene particles which contain embedded six different fluorochromes that align to the most common flow cytometry excitation and emission filter sets. The kit consists of five premixed populations of beads that are embedded with increasing amounts of the dye mixture. First, we determined the linearity of the fluorescence intensity signal in the FITC channel of the flow cytometer across the sensor dynamic range by comparing fluorescence intensities of each peak with the population MEFs (molecules of equivalent fluorochrome) as provided by the manufacturer (Supplementary Fig. 11a, b). These results indicated adequate linear response for reads between 1e3 and 2e5 (arbitrary units). These data also provided the

relative numbers of each bead population and the overall population average MEF per bead of 17,277. Next, we determined the equivalent MEF for FITC–dextran species used for delivery by comparing the fluorescence of the FITC–dextran species to the calibration beads using a spectrophotometer (SpectraMax iD3). Excitation and emission wavelengths of 485 nm and 535 nm, respectively, were selected to match the flow cytometer FITC channel excitation and emission band peaks (488 nm and 533 nm). The bead concentration was measured by a Nexcelom cell counter. The resulting measurements provided equivalences of 1 MEF = 3.5 attograms of 2000 kDa FITC–dextran, or 5.4 attograms of 4 kDa FITC–dextran. These equivalences, together with the calibration factors for fluorescence signal across increasing flow cytometer laser powers, allowed the flow cytometer fluorescence intensities to be converted to femtograms of FITC–dextran (Supplementary Fig. 14).

### Dextran localization and membrane imaging

For high resolution confocal imaging, Jurkat cells were stained with a membrane dye that covalently labels extracellular proteins following manufacturer instructions (MemBrite Fix 640/660, Biotium). Briefly, about 2 million Jurkat cells were incubated for 5 min at 37 °C with the pre-staining solution (included) then washed and incubated an additional 15 min on ice with the MemBrite dye. Cells were then washed into room temperature delivery solution (PBS with 0.5 mg/mL HA and 0.2 mM $CaCl_2$) containing 0.2 mg/mL 70 kDa FITC–dextran, processed through the chip, and immediately diluted tenfold in 3% paraformaldehyde. The fixed cells were diluted, allowed to settle onto coverglass, and imaged on a laser scanning confocal microscope at 60× magnification with an oil immersion objective. For imaging flow cytometry of membrane morphology (i.e., Supplementary Fig. 18), Jurkat cells were stained similarly with MemBrite Fix 405/430 (Biotium), processed through the chip, washed into fresh chilled PBS, and immediately imaged on the ImageStream MkII (i.e., without fixation).

### Statistics and reproducibility

All imaging flow cytometry data was analyzed using Amnis IDEAS®. All graphing and statistical analyses were performed with MATLAB 2019a. Statistical significance of differences in mean viability between conditions with different driving pressures was evaluated by one-way analysis of variance (ANOVA) followed by Tukey's honest significance difference correction for multiple comparisons. Plots of fluorescence intensity distributions (e.g., Fig. 4b, i–k and Fig. 5a) were generated using the Robust Statistical Toolbox for MATLAB (https://github.com/CPernet/Robust_Statistical_Toolbox). Specifically, the Random Average Shifted Histogram algorithm was used with automatic kernel estimation to generate the unbiased probability distributions from sampled fluorescence values. All underlying raw fluorescence values are provided in the Source Data file. No statistical method was used to predetermine sample size. No data were excluded from the analyses. Experiments were not randomized. The investigators were not blinded to allocation during experiments and outcome assessment.

### Reporting summary

Further information on research design is available in the Nature Portfolio Reporting Summary linked to this article.

## Data availability

Source data are provided with this paper. The raw videos, raw flow cytometry data, and full computational fluid dynamics solution datasets are large (>500 GB). Requests for these data should be addressed to the corresponding author and will be fulfilled in 2–4 weeks. Source data are provided with this paper.

## Code availability

The mesh, solver configuration, and fluid constitutive properties used for the computational fluid dynamics simulations have been made available at https://github.com/derinsevenler/viscoelastic-mechanoporation-rheotool-natcomm2023.

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

## Acknowledgements

This work was supported by the National Institute of Allergy and Infectious Disease (K99 AI167063, D.S.), the National Cancer Institute (R01 CA255602, M.T.), and the National Science Foundation Engineering Research Center on Advanced Technologies for the Preservation of Biological Systems (#1941543 M.T.). We thank Carlie Rein for assistance with microdevice fabrication, Shannon Stott and Ezgi Antmen for microscopy resources and assistance, and Jon Edd, Avanish Mishra and Kaustav Gopinathan for helpful discussion.

## Author contributions

D.S. and M.T. conceptualized the work, reviewed results, and reviewed the paper. D.S. designed and performed experiments and simulations, analyzed the data, and wrote the paper.

## Competing interests

D.S. and M.T. are inventors on a pending patent WO2022155186 "Viscoelastic mechanoporation systems and methods thereof" owned by the General Hospital Corporation (Boston, MA), which is related to microfluidic cell focusing and intracellular delivery technologies related to this work.
