## [Peer review file · Nature Communications]

Reviewers' comments:

Reviewer #1 (Remarks to the Author)

The authors reported a new approach for intracellular delivery and cell transfection using a spiral microfluidic device and viscoelastic solution. The scalability (i.e. cell processing rate $\sim 200\text{M}$ cells/min) is impressive and the design of the microchannel that can eliminate potential channel clogging is original.

However, to be considered as a novel innovative intracellular delivery platform should outperform current solutions in originality, high delivery/transfection level, high cell viability, high stability in cell functions, cost, throughput, primary cell validation, and more; however, it is believed that the current form does not meet these criteria much. Given that, the reviewer does not believe the manuscript is suitable to be published in Nature Communications. Please see below for more details.

1. Performance: the platform should have demonstrated they can do deliver and transfect "significantly" greater compared to not only current microfluidic mechanoporation solutions but also the state-of-the-art methods/approaches such as electroporation, LNP, and/or viral vectors.

No data was presented comparing their device results with electroporator(s), for instance Neon Transfection System or Nucleofector. Based on the used concentration (100 $\mu\text{g/ml}$) which is considered very high, the reviewer strongly believes the electroporator can easily beat the presented results particularly for Jurkats (check the Neon's results from ThermoFisher website).

On another note, the manuscript reads in a way that the authors' work is the best among cell mechanoporation methods by comparing theirs with Jarrell et al (Ref 35, 36); however, Ref. 35 and 36 does not fully represent microfluidic cell mechanoporation methods. Please tone down and give enough credits to other prior works, even for viral transduction in the introduction. No matter what, viral transduction is the winner as of today.

The authors described the other mechanoporation methods are inefficient, low viability, non-uniform or so (introduction and discussion sections). These statements are qualitative and incorrect. There are more works which can do a much better job than the presented research in delivery and transfection. Thus, please compare them in a quantitative manner.

2. Originality: The use of viscoelastic solution is one of the key claims of this work; however, it seems Ref. 26 have demonstrated this (a different viscoelastic solution was used but the overarching idea is the same). It should be noted that the authors did not explicitly refer to this work that should have been clearly stated.

3. Cell viability: Cell viability seems acceptable; however, a special solution was used. For instance, addition of Ca^{2+} ions or cytoplasmic buffer was used. This may increase the cell viability, but one of the main disadvantages of electroporation is the use of special buffer solutions which is not ideal from a user perspective, and it is costly. A desired method should use conventional cell media or PBS while achieving high cell transfection or delivery efficiency.

4. Cell stability: As the authors mentioned, no investigation on how the HA would affect the cell functionality. The reviewer strongly believes the polymer solution will be internalized into cells; thus, its effect should be investigated beyond cell viability. RNA sequencing is strongly recommended to validate its effect critically.

5. Primary cell validation: The authors tested primary T cells which is great; however only 70 kDa FITC-dextran was delivered. To be considered as a great delivery device, the authors should deliver mRNA, CRISPR-RNP to T cells and see they can get the same level of transfection or KO. Without

these results, it is under this work can be applied for cell-based research towards clinical trials at all.
Minor points

1. The cell shown in Fig. 1 center seems to be experiencing a symmetric extension which is not happening in the device. Please modify.
2. Explicitly show the mean fluorescence intensity fold change for the quantification of delivery. For example, compared to the control, 00 fold changes was recorded for a certain situation.
3. What is the final tested FITC-Dextran concentration? Please explicitly state. For example, 00 mg/mL.
4. The tested dextran size was inconsistent. Often 4, 70 or 2000 kDa were used. What happens in delivery results if 2000 kDa was delivered into primary T cells?
5. Normally how much of a total volume was injected into a single run? What is the maximum cell number that the current device can process without a failure?
6. What is the value of numerical analysis regarding the delivery efficiency?
7. What is the Figure 2g conclusion? What is the relationship among the cell area, aspect ratio, and delivery efficiency? Also, how long did it take to process?
8. What is the relationship between the tested HA solution and temperature? If tested at 20C (room) or 36C (body), do you expect different or similar results?
9. Some experiments were conducted only two times (Fig. 4 c and d?). Replicate at least three times.
10. HEK293T is known to be easy to transfect and deliver external biomolecules but the author's platform was not successful with PBS. Describe why.

Reviewer #2 (Remarks to the Author)

Title: High throughput intracellular delivery by viscoelastic mechanoporation
Authors: Sevenler and Toner

Review Summary

This manuscript presents a novel technique for delivering exogenous material into human cells. The viscoelastic technique has some advantages over other mechanical techniques, mainly process uniformity and the ability to avoid clogging since cells never touch a solid surface. Given the relatively high throughput the technique holds promise in an area desperate for therapeutically relevant alternatives. The authors do a thorough job of characterizing the technique with multiple payloads, including clinically relevant mRNA and RNPs. I would have liked to see more data in primary cells, but also recognize that this is a first publication on a new technique. In general I'm in support of publication if my questions/comments are sufficiently addressed.

Specific Comments

1. Page 4, paragraph 1. Are there any foreseeable regulatory challenges associated with using hyaluronic acid for engineered cell therapies? Are there any approved products that already use it as an additive?
2. One general question I have is how one can tune this system. Do you need to fabricate a new channel for each type of cell, or can one system be used for a range of cells by modulating the flow parameters? Further, how do all of the relevant physical parameters scale? All things being equal, if

the cell size decreases, how does that influence the necessary flow parameters?

3. There are several dimensionless groups mentioned in the manuscript including the Deborah number, Weissenberg number, and Elasticity number. It could be useful to have a paragraph, and perhaps a table, summarizing the relevant dimensionless groups and how they impact the physics of this transfection process in addition to the relevant operating range. Also, I'd be curious to know the Reynolds number range used in this process.

4. Page 4, last paragraph. Given that upstream focusing improves process consistency, it suggests that the cell location within the channel has a strong effect on the outcome. This makes sense. What does that then mean for the optimal cell concentration range for this process? Commercial static electroporation often uses very dense cell suspensions (e.g. 100-200M cells/ml), what is the limit here? I would imagine that the limitations would be set by how much the viscoelastic properties of the suspension change with concentration and the ability to ensure uniform/consistent conditions for the cells. I'm most interested in the concentration range that doesn't compromise performance.

5. Methods, Delivery experiments. Related to the above, the paper states that it can do hundreds of millions of cells per minute, but the concentration listed here is 3-10 million cells/ml. The flow rate appears to be around 2.4 ml/min, giving a throughput of more like 24 million cells/min rather than hundreds of millions of cells per minute. Have the authors actually demonstrated hundreds of millions of cells per minute?

6. Figure 4e. If I'm reading the figure properly, it seems that if you move beyond 2.7 ml/min the yield begins to decrease. Thus, though the system can operate at up to 4 ml/min it seems unlikely this would be used in practice since the efficiency or viability, or both seem to decrease.

7. Page 10, Paragraph 1. Am I reading this correctly, the delivery is better for larger molecules? I would expect that this method is driven by diffusive transport into the cell, intriguing that a larger molecule would be delivered better than a smaller one.

8. Page 10, Paragraph 3. The decreasing viability with mRNA delivery is curious, in work I've seen there hasn't been a major cytotoxic burden associated with mRNA.

9. Page 12, Paragraph 2. The buffer question is interesting. Is there any opportunity for this technique to use buffers that are closer to growth media? Electroporation is limited in that conductivity plays a major role in the process and is a factor in buffer selection, but this isn't the case with the present technique. If you could enable transfection without the need for a buffer exchange, or with minimal cell handling required, that would be a benefit from a process development perspective.

10. Page 14, Paragraph 1. Could the authors expand on this flow instability limitation further? I'm particularly interested in understanding the relevant operating window for this system.

11. Page 14, Paragraph 2. Is there any chance that the toxicity previously associated with mRNA delivery was actually due to the HA? This relates to an earlier question but is there a list of already approved substances that can result in a viscoelastic fluid? Just thinking about what happens if HA is a regulatory issue.

Minor Comments

Page 3, last paragraph. Change CRISPR-Ca9 to CRISPR-Cas9

Page 8, last paragraph. '100 million cells per minute' isn't a concentration, perhaps the authors meant per ml?

Reviewer #3 (Remarks to the Author)

Manuscript NCOMMS-23-12330-T entitled "High throughput intracellular delivery by viscoelastic mechanoporation" describes the combination of inertio-elastic cell focusing through a microfluidic contraction in a viscoelastic hyaluronic acid (HA) solution to increase the extensional viscosity of the solution to promote mechanical disruption of the cell membrane and exogenous solution material uptake into the cell through 'mechanoporation'. The mechanoporation process was first evaluated using a simple contraction/expansion type microdevice lacking flow focusing with 1 mg/mL HA demonstrating high efficiency delivery of 2000 kDa FITC labeled dextran into Jurkat cells but with

lower cell viability/recovery with increasing flow rates. A second iteration device used outward-spiraling microchannels to focus cells toward the outer wall of the spirals and ultimately centered within the extensional constriction. This device is also scalable, able to handle clinically relevant cell numbers up to 200 million cells/minute. The degree of cell extension and cell area was analyzed as a function of flow rate using extensional cytometry techniques. High efficiency delivery of both 4 and 2000 kDa FITC-dextran delivered to cells with a ~ 10 fold increase of the 4 kDa dextran compared to the larger 2000 kDa dextran. The work also showed that the amount of dextran uptake was dramatically decreased if the macromolecules were added following stretching (5-180 sec later). The work also demonstrated GFP encoding mRNA delivery and RNP delivery for gene editing using a synthetic crRNA sequence to knock out the T cell receptor in Jurkat cells, as well as 70 kDa FITC dextran to HEK 293T and primary T cells showing the versatility of the approach in terms of both cell types and delivery vectors.

This work is well described and thoroughly assesses the effect of HA concentration, other viscoelastic polymers (PEO), various cell types (Jurkat, HEK, and Primary T Cells) with a variety of delivered vectors (BSA, Dextran, mRNA, RNP, etc.), and the paper is clearly written and supported by supplementary materials. The design is also informed by full 3D computational fluid mechanics simulation of viscoelastic flows with appreciable inertia to assess the effect of increasing Deborah number and the deviatoric stress tensor to map the stress experienced by the cell as it transits through the constriction.

The weakness of the paper is that the whole study seems to be based on phenomenological observation/empirical optimization for the delivery with no insight into the mechanism for delivery. First, it is unclear if mechanoporation truly creates physical pores in the membrane versus promoting uptake via endocytic pathways. Since the study draws many parallels between mechanoporation and electroporation (EP), there is a wealth of evidence that some of the EP delivery is through endocytosis. In particular, there are several elegant studies by Yuan at Duke investigating the different endocytosis contributions to uptake (PLoS One. 2011;6(6):e20923. doi: 10.1371/journal.pone.0020923. Epub 2011 Jun 13, Mol Ther Methods Clin Dev. 2014 Dec 17;1:14058. doi: 10.1038/mtm.2014.58. eCollection 2014) in which clathrin-mediated endocytosis, caveolae, and pinocytosis pathways were knocked down to assess their contribution in EP mediated DNA transfection. Further, Thottacherry, et. al., demonstrated that the clathrin independent CLIC/GEEC (CG) pathway was rapidly activated following a stretch/relaxation response of adherent cells on a flexible membrane due to the rapid decrease in membrane tension and formation of membrane invaginations which promote uptake of material (Nat Commun 9, 4217 (2018). <https://doi.org/10.1038/s41467-018-06738-5>). This reference is entirely consistent with the results shown in Figure 4j,k since the GC activation timescale from stress relaxation is on the order of a few seconds or less. The manuscript would be markedly enhanced by examining these contributions to vector delivery.

The authors should also discuss the effect of not only extensional stress but application time on cell viability. Classic work by Hellums on red blood cell damage (Biophys J. 1972 Mar; 12(3): 257-273. doi: 10.1016/S0006-3495(72)86085-5) showed that RBC damage was a function of both shear stress as well as contact time (Hellums curve, Figure 8) with a threshold shear stress of ~ 1500 dynes/cm² at long timescales but could withstand much higher stresses for short contact times. Although Hellums' work was only for RBC damage, many of the ideas can be extrapolated to other cell types and may inform some of the viability results reported, especially for the initial device where cells were not focused and presumably some were exposed to higher shear stress at the constriction wall. The extensional stress experienced by the Jurkat/HEK/T cells in this work is much higher than 1500 dynes/cm² but the exposure time would be very low (especially at 200 million cells/min throughput). This exposure time should be reported and could be compared with Hellums curve for use as a design guide.

Minor concerns:

1) Reference 35 and 36 describe work by Indee labs which is trying to commercialize mechanical DNA

uptake via vortex shedding devices; perhaps mention this commercialization effort in the text.

2) Results Paragraph 2, "Across tested all geometries..." is awkward, rephrase.

3) Figure 2, and Methods Rigid microfluidic device fabrication and fixturing. In the paper (and in the Figure 2 caption) it is unclear that the device is fabricated via epoxy molding rather than from PDMS. Please make this clear. Also, although reference 50 is cited please provide the epoxy used to cast the device (EpoxAcast™ 690 transparent epoxy kit?)

4) Figure S4, unclear what "'0 bar' indicates corresponding unstretched controls" is referring to. There is no '0 bar' label in legend.

5) Figure S11, S11c is not captioned.

6) Figure S20 is not referenced in text. It is referenced in the mechanical analysis on the following page of the SI file.

We are submitting our response to the criticisms raised by the referees to our article entitled “High throughput intracellular delivery by viscoelastic mechanoporation.” We would like to thank the Reviewers for their valuable comments, and we found the criticisms very helpful in clarifying important aspects of our work. The following changes have been incorporated into a revised version of the manuscript.

Response to Review 1

The authors reported a new approach for intracellular delivery and cell transfection using a spiral microfluidic device and viscoelastic solution. The scalability (i.e. cell processing rate ~ 200M cells/min) is impressive and the design of the microchannel that can eliminate potential channel clogging is original. However, to be considered as a novel innovative intracellular delivery platform should outperform current solutions in originality, high delivery/transfection level, high cell viability, high stability in cell functions, cost, throughput, primary cell validation, and more; however, it is believed that the current form does not meet these criteria much.

We are grateful to the Reviewer’s positive comments and constructive feedback. As detailed below, we have revised the manuscript to highlight how viscoelastic mechanoporation does indeed outperform current solutions in originality, high delivery/transfection level, high cell viability, cost, throughput, and more. We have also added more data, tables, and supplemental figures to strengthen the manuscript according to the Reviewer’s suggestions. Please see responses below for more details to Reviewer’s specific questions and comments, and where exactly in the manuscript the recommended changes have been made.

1. Performance: the platform should have demonstrated they can do deliver and transfect "significantly" greater compared to not only current microfluidic mechanoporation solutions but also the state-of-the-art methods/approaches such as electroporation, LNP, and/or viral vectors. No data was presented comparing their device results with electroporator(s), for instance Neon Transfection System or Nucleofector. Based on the used concentration (100 ug/ml) which is considered very high, the reviewer strongly believes the electroporator can easily beat the presented results particularly for Jurkats (check the Neon’s results from Thermofisher website). We agree with the Reviewer that quantitative comparisons with other membrane disruption technologies. We have included a supplementary table providing quantitative comparison to show that viscoelastic mechanoporation is a significant improvement compared to all existing and emerging direct delivery technologies in the literature in terms of transfection throughput, efficiency, and cost due to the use of high cell concentrations (Supplemental page 5, lines 61-91). Data provided in the manuscript clearly demonstrate: viscoelastic mechanoporation is by far the highest throughput microfluidic intracellular delivery method reported to date, while also demonstrating nearly perfect transfection performance (>90% transfection efficiency of Jurkat cells with statistically insignificant cell death or cell loss), all the while using higher cell concentrations than all other microfluidic methods. Regarding viral vectors, LNP delivery and conventional bulk electroporation, they have major intrinsic limitations as we stated in the introduction (page 2, lines 27-31 and 38-41): “Viral vectors such as lentivirus are a well-

established platform technology that is clinically approved for *ex vivo* genetic manipulation of immune cells, however they are unable to target a specific genetic locus, have a limited genetic payload, and can be expensive to produce themselves⁴⁻⁶. Synthetic vector systems such as DNA-cationic polymer complexes are a scalable alternative used in bio-manufacturing but can be unstable, inefficient, and/or cytotoxic^{7,8}... Scaled-up electroporation schemes face additional challenges associated with nonuniform E-fields, heating, electrolysis, pH changes, electrode corrosion, ionic contamination, and prolonged exposure of cells to low-conductance electroporation buffer⁹.”

Based on the used concentration (100 ug/ml) which is considered very high, the reviewer strongly believes the electroporator can easily beat the presented results particularly for Jurkats (check the Neon’s results from Thermofisher website).

Regarding mRNA concentration, 100 µg/mL is in fact well within the typical range used for most electroporation and mechanoporation recipes. For example, the Neon system the Reviewer mentions recommends mRNA concentrations in the range of 100 µg/mL to 150 µg/mL, shown in the Neon application notes available online (<https://www.thermofisher.com/order/catalog/product/NEON1> and <http://www.thermofisher.com/transfectionprotocolsandcitations>). 100 µg/mL is also typical when compared to emerging high-throughput electroporation (Sido et al, 2021: 200 µg/mL, Lissandrello et al, 2020: 50 µg/mL) and mechanoporation (Loo et al, 2021: 40 µg/mL) approaches. We have revised the mRNA results section to make this clear with these references (page 11, lines 203-205).

On another note, the manuscript reads in a way that the authors’ work is the best among cell mechanoporation methods by comparing theirs with Jarrell et al (Ref 35, 36); however, Ref. 35 and 36 does not fully represent microfluidic cell mechanoporation methods. Please tone down and give enough credits to other prior works, even for viral transduction in the introduction. No matter what, viral transduction is the winner as of today.

The authors described the other mechanoporation methods are inefficient, low viability, non-uniform or so (introduction and discussion sections). These statements are qualitative and incorrect. There are more works which can do a much better job than the presented research in delivery and transfection. Thus, please compare them in a quantitative manner.

We agree with the Reviewer that a direct and comparison between methods would strengthen the manuscript. We have included a table of quantitative comparison of the efficiency, throughput, and cell concentration used in all the emerging mechanoporation and electroporation delivery technologies in the literature (Supplemental page 5, lines 61-91). We likewise agree with the Reviewer that viral vectors are a useful tool for cell and gene therapy manufacturing, and that prior studies on mechanoporation have made major contributions that deserve to be acknowledged. We have toned down the manuscript with more credit and discussion of the impact of prior work (page 2, lines 27-31, 50-55 and page 14, lines 264-269 and page 15, lines 281-286).

2. Originality: The use of viscoelastic solution is one of the key claims of this work; however, it seems Ref. 26 have demonstrated this (a different viscoelastic solution was used but the

overarching idea is the same). It should be noted that the authors did not explicitly refer to this work that should have been clearly stated.

We revised the manuscript to explicitly refer to Ref. 26 as suggested as well as clearly described the novelty of our work compared to Ref. 26 (pages 14-15, lines 277-286 and Supplemental pages 2-3, lines 19-30). Briefly, Ref. 26 does not show or use viscoelastic normal stress differences to achieve poration and as a result cells contact with the channel walls to impart mechanical poration. In addition, in Ref. 26 the solution was explicitly described and modeled as a generalized Newtonian (shear thinning) solution, not as a viscoelastic solution. Although methylcellulose (MC) solutions have some elasticity and Ref. 26 used the word ‘viscoelastic’ to describe the MC solution in several places, it never discussed any viscoelastic property of MC nor was it discussed how viscoelasticity was relevant to mechanoporation. None of the fundamental concepts of viscoelasticity such as ‘normal stress’, ‘relaxation time,’ ‘strain hardening,’ or ‘extensional flow’ appear in Ref. 26. In fact, in Ref. 26, mechanoporation required a channel that was significantly smaller than the cells (optimal width of 12 μm , with 16 μm diameter K562 cells, decreased to 10 μm , for 13 μm diameter Jurkat cells). In Figure 2F and in the text of Ref. 26, the delivery efficiency and amount of delivered material decreased as the gap width was increased even slightly (yet still smaller than the cells), suggesting that wall contact was required for mechanoporation in a manner similar to other approaches which also used narrow microfluidic contractions in a comparable range (i.e., Sharei et al, *PNAS* 2013).

In addition to differences in the mechanism of poration between Ref. 26 and our study, there are also significant operational differences. In Ref. 26, the mechanoporation throughput was nearly three orders of magnitude lower than this work (at best 0.3 million cells per minute vs. over 200 million cells per minute in this work) and the device was highly susceptible to clogging. The device included an integrated filter that became clogged with lysed cells after just 0.5-5 million cells in total, and clogging was exacerbated with increasing cell concentrations up to 5 million cells per mL (text and Figure S4-S6). This is about the same rate of clogging as prior articles on mechanical cell squeezing (i.e., Ding et al, *Nature Biomedical Engineering* 2017). In addition, in Ref. 26, the transfection performance was highly nonuniform. Even after comprehensive optimization of the microfluidic geometry and delivery solution, about half of the recovered cells died. Notably, Ref. 26 did not report the number of recovered cells, so the significant fraction of cells that were retained or lysed by the device were neglected from the analysis, and 50% viability represented an optimistic upper bound. In our study, cell yield and viability were included in the reported data with both delivery efficiency and viability exceeding 90%. We revised the manuscript to make these differences explicitly clear (pages 14-15, lines 279-284 and Supplemental pages 2-3, lines 19-30).

3. Cell viability: Cell viability seems acceptable; however, a special solution was used. For instance, addition of Ca^{2+} ions or cytoplasmic buffer was used. This may increase the cell viability, but one of the main disadvantages of electroporation is the use of special buffer solutions which is not ideal from a user perspective, and it is costly. A desired method should use conventional cell media or PBS while achieving high cell transfection or delivery efficiency. As with electroporation, the cost of solution components is negligible compared to the cost of the biomolecule cargo. For example, for a typical 200 μL sample, the cost of the hyaluronic acid component was at most \$0.06 (e.g., 0.4 mg at \$0.150/mg) while the cost of mRNA was about

\$80 (e.g., 100 $\mu\text{g}/\text{mL}$ at $\$4/\mu\text{g}$) and CRISPR-Cas9 RNP was about \$12 (e.g., 8 μg at $\$1.50/\mu\text{g}$ for the Cas9 protein alone). Likewise, both the conventional sodium phosphate buffered saline and ‘cytoplasmic’ potassium solutions were compositionally simpler and less expensive than culture medium. Additives (such as ATP and glucose) that can improve the cell recovery are widely used by many membrane disruption technologies (including but not limited to electroporation-based methods). We revised the manuscript to address these points (page 11, lines 221-222) as well as included a new supplemental table with this cost comparison (Table S2, “Reagent cost breakdown”, Supplemental page 8, lines 101-103).

4. Cell stability: As the authors mentioned, no investigation on how the HA would affect the cell functionality. The reviewer strongly believes the polymer solution will be internalized into cells; thus, its effect should be investigated beyond cell viability. RNA sequencing is strongly recommended to validate its effect critically.

We agree that RNA sequencing could be valuable for determining the impacts of the process on cell phenotype, however this level of analysis will be suited for future work towards specific biomedical applications and is beyond the scope of this article. However, we provided direct evidence that viscoelastic mechanoporation is also compatible with other polymers such as polyethylene oxide and not just HA, so the best choice of polymer should be selected for the intended application and the cell type used (pages 10-11, lines 196-198 and page 15, lines 309-311).

5. Primary cell validation: The authors tested primary T cells which is great; however only 70 kDa FITC-dextran was delivered. To be considered as a great delivery device, the authors should deliver mRNA, CRISPR-RNP to T cells and see they can get the same level of transfection or KO. Without these results, it is under this work can be applied for cell-based research towards clinical trials at all.

We appreciate the Reviewer’s encouragement and share their enthusiasm for applications in therapeutic T cell engineering. The objective of this article is to introduce the fundamental approach of viscoelastic mechanoporation and show it is applicable to a variety of potential cell types and high throughput applications in cell and gene therapy production. Therefore, optimization and detailed phenotypic studies of this approach for specific applications such will be more appropriate in a separate follow up article.

Minor points

1. The cell shown in Fig. 1 center seems to be experiencing a symmetric extension which is not happening in the device. Please modify.

Based on the experimental and computational studies of flow kinematics, the dominant contributor to the fluid internal stresses along the center streamline was found to be the uniaxial extensional component. Furthermore, the shape of the stretched cells was approximately front-back (‘leading-trailing’) symmetric (Figure 2f-g). Therefore, the conceptual schematic showing a cell in a uniaxial extensional flow undergoing a symmetric extension is well-justified, both from an understanding of the fluid kinematics as well as the experimentally observed cell shape.

2. Explicitly show the mean fluorescence intensity fold change for the quantification of delivery. For example, compared to the control, 00 fold changes was recorded for a certain situation.

In this paper, we conducted quantitative fluorescence cytometry to measure the absolute mass of delivered material, for a range of molecular weight molecules and extracellular concentrations of those molecules (Figures 4i, S11, and S12). This is the gold-standard approach for quantifying delivery, and it is a more meaningful metric than fold-change fluorescence intensity. Fold-change intensity is affected by all the sources of background (stray light, amplifier dark current, and autofluorescence) which vary between imaging systems and sample type, as well as the labeling density of the fluorophore on the dye molecule. We revised the manuscript to make this point clear (page 10, lines 176-179).

3. What is the final tested FITC-Dextran concentration? Please explicitly state. For example, 00 mg/mL.

We included this information in the Methods subsection “Delivery Experiments”, namely “Except where stated otherwise...0.2 mg/mL 70 kDa FITC-dextran.” We have also added this throughout the Results section as well to make this clearer (page 4, lines 80-86 and page 8, lines 151-152).

4. The tested dextran size was inconsistent. Often 4, 70 or 2000 kDa were used. What happens in delivery results if 2000 kDa was delivered into primary T cells?

We revised the manuscript to remove any ambiguities about dextran sizes used in different experiments (page 4, lines 80-86 and page 8, lines 151-152). Briefly, FITC-dextran with a molecular weight of 70 kDa was used consistently for all the cell types tested with the final device (Jurkat cells, HEK293T cells, and primary T cells, Figures 4 and 5). 4 kDa and 2,000 kDa FITC-dextran were used in a preliminary study and where the relationships between cargo molecule size and delivery kinetics were of primary interest (Figures 4i-k).

5. Normally how much of a total volume was injected into a single run? What is the maximum cell number that the current device can process without a failure?

Regarding the volume used for each run, we include this information in the Methods subsection “Delivery Experiments,” and state “200 μ L samples were typically processed within about 5 seconds” (page 18, lines 360-361). Regarding the maximum cell number, the revised manuscript now contains new supplemental data demonstrating efficient delivery to about 10x larger samples (25 million cells per sample, Figure S12 and page 10, lines 171-173).

6. What is the value of numerical analysis regarding the delivery efficiency?

These numerical simulations showed that viscoelastic normal stress differences were more dominant than shear stress within the flow, especially along the channel centerline where cells were focused. This supported the hypothesis that the fluid viscoelasticity was required for efficient and uniform mechanoporation, which was also shown in the experimental observation that the delivery efficiency was much less uniform if the polymer was not used.

7. What is the Figure 2g conclusion? What is the relationship among the cell area, aspect ratio, and delivery efficiency? Also, how long did it take to process?

Briefly, regarding image processing time, manual annotation of several thousand frames took several days of work (perhaps 10 hours or so), model training took about 18 hours, and automated image segmentation took roughly 1 second per image (i.e., about 30 hours for the entire data set). We have revised the manuscript to include this information (page 17, lines 341-

342). Regarding the cell deformation measurements in Figure 2g, in the Results subsection “High throughput cell focusing enables high throughput and consistent cell deformation”, we state “As expected, cells were elongated along the flow direction and cell aspect ratio increased with increasing flow rate. Qualitatively, at lower strain rates cells remained rounded, while at higher strain rates cells were transiently pulled into an elongated spindle-like morphology, consistent with previous studies of transient cell deformation in pure extensional flows [22,32,33]” (page 6, lines 111-115).

8. What is the relationship between the tested HA solution and temperature? If tested at 20C (room) or 36C (body), do you expect different or similar results?

Based on the time-temperature superposition principle, the fluid relaxation time and polymeric contribution to viscosity can both be expected to decrease with increasing temperature. However, the deformation is far outside the range of linear elasticity theory, so the extent of the decrease would need to be measured experimentally.

9. Some experiments were conducted only two times (Fig. 4 c and d?). Replicate at least three times.

We agree that n=3 replicates strengthens the validation study. We repeated the experiments for Figures 4 b, c, d and e with n=3 replicates and updated the figures, adding an additional supplemental Figure S10 for relative cell recovery. The additional replicates agreed with the previous data and did not change the results. These data were also subjected to significance analysis by 1-way ANOVA with Tukey’s honest significant difference correction for multiple comparisons.

10. HEK293T is known to be easy to transfect and deliver external biomolecules but the author’s platform was not successful with PBS. Describe why.

The Reviewer is correct that HEK293T is known as an ‘easy to transfect’ cell type, but this is mainly due to a tolerance for plasmid delivery, especially with cationic polymers such as polyethyleneimine. Compared with the suspension cell line that was evaluated (Jurkat cells), HEK293T cells were substantially larger (16 μm), and as an adherent line they can have different cytoskeletal mechanical properties. HEK293T cells may also have had a different amount of excess (i.e., ruffled) plasma membrane with respect to their size. This list is not meant to be exhaustive but provide a few examples of biological differences which can contribute to the magnitude and duration of membrane tension that was applied, as well as to the different response of the cell to that stress history, that might have resulted in a different distribution in pores than were created in Jurkat cells under similar flow conditions. We revised the manuscript to include these points (page 13 lines 234-236).

Response to Review 2

Review Summary

This manuscript presents a novel technique for delivering exogenous material into human cells. The viscoelastic technique has some advantages over other mechanical techniques, mainly process uniformity and the ability to avoid clogging since cells never touch a solid surface. Given the relatively high throughput the technique holds promise in an area desperate for therapeutically relevant alternatives. The authors do a thorough job of characterizing the

technique with multiple payloads, including clinically relevant mRNA and RNPs. I would have liked to see more data in primary cells, but also recognize that this is a first publication on a new technique. In general I'm in support of publication if my questions/comments are sufficiently addressed.

We are grateful to the Reviewer for the supportive comments. Detailed point-by-point responses to comments are provided below.

Specific Comments

1. Page 4, paragraph 1. Are there any foreseeable regulatory challenges associated with using hyaluronic acid for engineered cell therapies? Are there any approved products that already use it as an additive?

We are grateful to the Reviewer for raising this relevant concern. Hyaluronic acid is considered safe to use for several therapeutic applications including visco-supplementation (intra-articular injection of HA to joints to alleviate osteoarthritis). However, we showed in the manuscript that viscoelastic mechanoporation is compatible with other biocompatible polymers. For applications that involve transfection of cells that will be introduced into the patient, selection of an appropriate and safe biocompatible polymer will be required. We have revised the Discussion to address this point briefly (page 16 lines 309-312).

2. One general question I have is how one can tune this system. Do you need to fabricate a new channel for each type of cell, or can one system be used for a range of cells by modulating the flow parameters? Further, how do all of the relevant physical parameters scale? All things being equal, if the cell size decreases, how does that influence the necessary flow parameters?

We thank the Reviewer for these clarifying questions. We demonstrated that a single microfluidic geometry could be used to efficiently deliver dextran to Jurkat cells, HEK cells, and activated primary T cells. We also showed the buffer composition could also be optimized for different cell types. We made this point clear in the revised manuscript. Regarding scaling laws with cell size, we did not observe stratification in the delivery efficiency or viability with cell size within a given sample. We have noted these in the Discussion section (page 14 lines 254-258).

3. There are several dimensionless groups mentioned in the manuscript including the Deborah number, Weissenberg number, and Elasticity number. It could be useful to have a paragraph, and perhaps a table, summarizing the relevant dimensionless groups and how they impact the physics of this transfection process in addition to the relevant operating range. Also, I'd be curious to know the Reynolds number range used in this process.

We agree that this discussion would be helpful and have added a new supplemental appendix, "Dimensionless numbers and their approximate values for viscoelastic mechanoporation" (Supplemental page 2-4, lines 10-58).

4. Page 4, last paragraph. Given that upstream focusing improves process consistency, it suggests that the cell location within the channel has a strong effect on the outcome. This makes sense. What does that then mean for the optimal cell concentration range for this process? Commercial static electroporation often uses very dense cell suspensions (e.g. 100-200M cells/ml), what is the limit here? I would imagine that the limitations would be set by how much the viscoelastic properties of the suspension change with concentration and the ability to ensure

uniform/consistent conditions for the cells. I'm most interested in the concentration range that doesn't compromise performance.

In Figure 4h and the Results subsection "Optimization of viscoelastic mechanoporation for high throughput and uniform intracellular delivery," we showed that the transfection performance did not decrease for cell concentrations up to 100 million cells per mL (page 10, lines 168-171).

5. Methods, Delivery experiments. Related to the above, the paper states that it can do hundreds of millions of cells per minute, but the concentration listed here is 3-10 million cells/ml. The flow rate appears to be around 2.4 ml/min, giving a throughput of more like 24 million cells/min rather than hundreds of millions of cells per minute. Have the authors actually demonstrated hundreds of millions of cells per minute?

We thank the Reviewer for finding a typographic error in that section, discussed below (100 million cells per mL was mistakenly written as 100 million cells per minute, this has been corrected, also discussed in the minor comments below) and apologize for any confusion caused by the typo.

6. Figure 4e. If I'm reading the figure properly, it seems that if you move beyond 2.7 ml/min the yield begins to decrease. Thus, though the system can operate at up to 4 ml/min it seems unlikely this would be used in practice since the efficiency or viability, or both seem to decrease.

The Reviewer is correct, the viability decreases above 2.7 mL/min for Jurkat cells. In the article, 4 mL/min was the highest tested flow rate during the optimization studies for Jurkat cells.

7. Page 10, Paragraph 1. Am I reading this correctly, the delivery is better for larger molecules? I would expect that this method is driven by diffusive transport into the cell, intriguing that a larger molecule would be delivered better than a smaller one.

We clarified in the revised manuscript that the delivery was more efficient for smaller molecules than for larger ones, as the Reviewer expected: "Across all extracellular concentrations, the average amount of 4 kDa dextran delivered per cell was roughly 10-fold higher than for 2,000 kDa dextran." We have revised this sentence to make it more clear (page 10, lines 179-181).

8. Page 10, Paragraph 3. The decreasing viability with mRNA delivery is curious, in work I've seen there hasn't been a major cytotoxic burden associated with mRNA.

We agree with the Reviewer that cytotoxicity following mRNA delivery was unexpected. The data seemed to show a gentle trend towards increasing cell death with increasing mRNA concentration. One possible mechanism is that mRNA is itself a linear polymer, so increased concentrations of mRNA would increase the elasticity of the solution. This would result in higher membrane tension (i.e., beyond optimal) at a given flow rate. We have revised the text to include this potential mechanism (page 11, lines 208-211).

9. Page 12, Paragraph 2. The buffer question is interesting. Is there any opportunity for this technique to use buffers that are closer to growth media? Electroporation is limited in that conductivity plays a major role in the process and is a factor in buffer selection, but this isn't the case with the present technique. If you could enable transfection without the need for a buffer exchange, or with minimal cell handling required, that would be a benefit from a process development perspective.

We thank the Reviewer for this insight. We have added mention of this idea to the Discussion section (page 14, 257-258).

10. Page 14, Paragraph 1. Could the authors expand on this flow instability limitation further? I'm particularly interested in understanding the relevant operating window for this system.
We agree that further explanation would be helpful, and we have expanded this section of the Discussion (page 15, line 299-308). Briefly, the flow instability did not inhibit efficient and gentle mechanoporation, perhaps because peak stresses are expected to be limited to the narrowest portion of the contraction

11. Page 14, Paragraph 2. Is there any chance that the toxicity previously associated with mRNA delivery was actually due to the HA? This relates to an earlier question but is there a list of already approved substances that can result in a viscoelastic fluid? Just thinking about what happens if HA is a regulatory issue.
Please see response to comment 8 above.

Minor Comments

Page 3, last paragraph. Change CRISPR-Ca9 to CRISPR-Cas9

Page 8, last paragraph. '100 million cells per minute' isn't a concentration, perhaps the authors meant per ml?

The Reviewer is correct on both counts, we have corrected these in the revision.

Response to Reviewer 3

This work is well described and thoroughly assesses the effect of HA concentration, other viscoelastic polymers (PEO), various cell types (Jurkat, HEK, and Primary T Cells) with a variety of delivered vectors (BSA, Dextran, mRNA, RNP, etc.), and the paper is clearly written and supported by supplementary materials. The design is also informed by full 3D computational fluid mechanics simulation of viscoelastic flows with appreciable inertia to assess the effect of increasing Deborah number and the deviatoric stress tensor to map the stress experienced by the cell as it transits through the constriction.

The weakness of the paper is that the whole study seems to be based on phenomenological observation/empirical optimization for the delivery with no insight into the mechanism for delivery. First, it is unclear if mechanoporation truly creates physical pores in the membrane versus promoting uptake via endocytic pathways. Since the study draws many parallels between mechanoporation and electroporation (EP), there is a wealth of evidence that some of the EP delivery is through endocytosis. In particular, there are several elegant studies by Yuan at Duke investigating the different endocytosis contributions to uptake (PLoS One. 2011;6(6):e20923. doi: 10.1371/journal.pone.0020923. Epub 2011 Jun 13, Mol Ther Methods Clin Dev. 2014 Dec 17;1:14058. doi: 10.1038/mtm.2014.58. eCollection 2014) in which clathrin-mediated endocytosis, caveolae, and pinocytosis pathways were knocked down to assess their contribution in EP mediated DNA transfection. Further, Thottacherry, et. al., demonstrated that the clathrin independent CLIC/GEEC (CG) pathway was rapidly activated following a stretch/relaxation response of adherent cells on a flexible membrane due to the rapid decrease in membrane tension and formation of membrane invaginations which promote uptake of material (Nat

Commun 9, 4217 (2018). <https://doi.org/10.1038/s41467-018-06738-5>). This reference is entirely consistent with the results shown in Figure 4j,k since the GC activation timescale from stress relaxation is on the order of a few seconds or less. The manuscript would be markedly enhanced by examining these contributions to vector delivery.

We thank the reviewer for including these informative and helpful references. The reviewer is correct that endocytosis is well-established as the primary mechanism for large biomolecule delivery in EP. To address this concern, we have included new data from additional experiments to address this question, and also added new discussion along these lines in the context of the references included by the reviewer (page 15 lines 287-293). Briefly, all the results are in agreement with the hypothesis that biomolecules are delivered to the cytosol via mechanopores (and also consistent with prior work by others on mechanoporation) while several of the results are in conflict with the hypothesis that delivery is primarily mediated by endocytosis. Here we summarize our evidence in conflict with endocytosis as a primary mechanism:

- While membrane stretching does promote endocytosis and material uptake, the effect is moderate (i.e. doubling the baseline rate of endocytosis, Thottacherry et al, Nature Communications 2018). In contrast, we showed over hundredfold greater delivery compared to unstretched (endocytosis only) controls (Figure 4b and 4i). Indeed, 4 kDa FITC-dextran was approximately equilibrated between the intracellular and extracellular compartments (Figure S14 and page 10 lines 181-183).
- Endocytosis decreases significantly at sub-normothermic temperatures (i.e. room temperature, Thottacherry et al, Nature Communications 2018, Figure 4). In our studies, all experiments were performed at room temperature, and the delivery solution was at room temperature (membrane staining was performed on ice).
- In a new experiment, the outer cell membrane was covalently labeled with a fluorescent molecule, then cells were immediately mixed with a delivery solution containing 70 kDa FITC-dextran, processed through the chip, immediately fixed with paraformaldehyde, and imaged by confocal microscopy. Endocytosis was minimized by ensuring cells were exposed to the delivery solution as little as possible (less than 5 minutes), and by ensuring cells were never warmed to 37C after being labeled (on ice) then transfected and immediately fixed (at room temperature). 60x confocal images showed FITC-dextran throughout the cytosol, rather than localized to any endosomes which retained the covalent membrane dye (Figure S15, page 10 lines 188-193 and page 20 lines 427-434).

The authors should also discuss the effect of not only extensional stress but application time on cell viability. Classic work by Hellums on red blood cell damage (Biophys J. 1972 Mar; 12(3): 257–273. doi: 10.1016/S0006-3495(72)86085-5) showed that RBC damage was a function of both shear stress as well as contact time (Hellums curve, Figure 8) with a threshold shear stress of ~ 1500 dynes/cm² at long timescales but could withstand much higher stresses for short contact times. Although Hellums' work was only for RBC damage, many of the ideas can be extrapolated to other cell types and may inform some of the viability results reported, especially for the initial device where cells were not focused and presumably some were exposed to higher shear stress at the constriction wall. The extensional stress experienced by the Jurkat/HEK/T cells in this work is much higher than 1500 dynes/cm² but the exposure time would be very low

(especially at 200 million cells/min throughput). This exposure time should be reported and could be compared with Hellums curve for use as a design guide.

We thank the Reviewer for drawing this interesting work to our attention. We have added a discussion that contextualizes the timescales of peak stress in this work (about 10 μ sec) compared with the article described by the Reviewer (page 15 lines 299-304). We thought it pertinent to also mention as part of that discussion Evan Evan's seminal work on dynamic tension spectroscopy, wherein he showed membranes could withstand much higher tensions on shorter timescales and provided a biophysical explanation based on rates of defect formation and growth in tensioned membranes.

Minor concerns:

1) Reference 35 and 36 describe work by Indee labs which is trying to commercialize mechanical DNA uptake via vortex shedding devices; perhaps mention this commercialization effort in the text.

We thank the Reviewer for highlighting this important work. We are familiar with the impressive contributions from Indee Labs which we referenced but discussing any commercialization efforts in an academic publication may represent a conflict of interest.

2) Results Paragraph 2, "Across tested all geometries..." is awkward, rephrase.

We thank the Reviewer for the close read, this typo has been fixed.

3) Figure 2, and Methods Rigid microfluidic device fabrication and fixturing. In the paper (and in the Figure 2 caption) it is unclear that the device is fabricated via epoxy molding rather than from PDMS. Please make this clear. Also, although reference 50 is cited please provide the epoxy used to cast the device (EpoxAcast™ 690 transparent epoxy kit?)

We have added mention of the epoxy device to the Figure 2 caption and added substantially more information about the device fabrication protocol (page 17 lines 322-336).

4) Figure S4, unclear what "'0 bar' indicates corresponding unstretched controls" is referring to. There is no '0 bar' label in legend.

We have revised this sentence to clarify that these controls were incubated with the dextran solution but not pumped through the chip (Supplemental page 12, Figure S4).

5) Figure S11, S11c is not captioned.

We thank the Reviewer for pointing out this oversight. We have added a caption entry for the table in Figure S11c (now Figure S13c, Supplemental page 21).

6) Figure S20 is not referenced in text. It is referenced in the mechanical analysis on the following page of the SI file.

We have moved the supplementary discussion section to the beginning of the Supplemental information section so the reader encounters this discussion before Figure S20 (now Figure S24).

REVIEWER COMMENTS

Reviewer #1 (Remarks to the Author):

The authors have diligently addressed the reviewer's prior feedback; however, a few concerns still remain as follows.

1. Performance Evaluation: While Table S1 effectively facilitates a comparison with prevailing microfluidics-based delivery systems, it is imperative to juxtapose the performance of the proposed device against an electroporator (such as Neon or Nucleofector) using equivalent mRNA concentrations. Conducting this experiment would not be overly demanding, and the reviewer is particularly intrigued by the transfection efficiency of the device in the context of primary T cells. This aspect carries significance as electroporation is the prevalent method in most foundational laboratories. It is thus crucial for the proposed device to showcase its superior performance in comparison.
2. Cellular Stability Assessment: While the reviewer acknowledges the authors' response, given the submission to a high-impact journal, there arises the necessity for genomic functional profiling. This step should be undertaken to strengthen the manuscript's scientific rigor and contribute to its appropriateness for publication in a journal of this caliber.
3. Validation with Primary Cells: The absence of primary cell validation would potentially make the manuscript to the realm of conceptual exploration, devoid of tangible impact. The reviewer strongly advocates for the execution of primary CRISPR tests as an essential validation step to substantiate the device's utility and potency.

Reviewer #2 (Remarks to the Author):

I have reviewed the author comments and the updated manuscript and feel that the current version addresses my concerns sufficiently.

Reviewer #3 (Remarks to the Author):

NCOMMS-23-12330A is a revision of the manuscript entitled "High throughput intracellular delivery by viscoelastic mechanoporation". Overall, the authors were very responsive to the concerns raised by all reviewers and revised the manuscript appropriately to address these concerns. The manuscript was revised to include more clarity about HA and vector concentrations, volume processed, and appropriate inertio-elastic stretching times as well as including a third replicate of some experiments. In particular, the revision addresses reviewer 1's concerns about originality, performance, and cost especially in the included tables in the supplementary information file. The revision also addressed this reviewer's major concerns about delivery mechanisms, showing confocal imaging of Jurkat cells with the cell membrane dyed prior to and following viscoelastic mechanoporation and FITC-Dextran delivery. Following mechanoporation the FITC-Dextran is found to be relatively uniformly distributed throughout the cytoplasm. When examining the images in Figure S15 two things stand out to me: 1) The membrane morphology following viscoelastic mechanoporation looks different with a number of highly localized dyed puncta around the cell. It is unclear if this is due to membrane relaxation and decrease in cell volume while preserving membrane area. 2) The distribution of FITC fluorescence also shows some variation of fluorescence intensity throughout the cytoplasm with a speckled appearance and some distinct areas of higher fluorescence. It is similarly unclear if this is due to structural differences throughout the cytosol (e.g., organelle distribution). Since only 2 images are presented, no

real conclusions can be drawn, but it may be interesting to explore secondary, longer term, delivery mechanisms especially for larger vectors like plasmid DNA (MW >1M kDa).

Response to Review 1

The authors have diligently addressed the reviewer's prior feedback; however, a few concerns still remain as follows.

1. Performance Evaluation: While Table S1 effectively facilitates a comparison with prevailing microfluidics-based delivery systems, it is imperative to juxtapose the performance of the proposed device against an electroporator (such as Neon or Nucleofector) using equivalent mRNA concentrations. Conducting this experiment would not be overly demanding, and the reviewer is particularly intrigued by the transfection efficiency of the device in the context of primary T cells. This aspect carries significance as electroporation is the prevalent method in most foundational laboratories. It is thus crucial for the proposed device to showcase its superior performance in comparison.

We agree with the Reviewer that comparison with electroporation is valuable because it is widely used in laboratories for RNA delivery. On the other hand, there is now substantial evidence that laboratory-scale electroporation results are not directly translatable to high throughput electroporation processes, as we describe (manuscript page 2, lines 49-51). We were able to find two highly relevant and recent studies from two different groups that performed head-to-head comparisons of electroporation and flow-based mechanoporation to answer these specific questions, both in terms of delivery and cell function at the RNA gene expression and protein levels. We have added a new discussion section that contextualizes our results based on the findings of these two relevant studies (**page 16, lines 298-306**).

2. Cellular Stability Assessment: While the reviewer acknowledges the authors' response, given the submission to a high-impact journal, there arises the necessity for genomic functional profiling. This step should be undertaken to strengthen the manuscript's scientific rigor and contribute to its appropriateness for publication in a journal of this caliber.

As much as we agree with the reviewer that genomic functional profiling is useful for comparing the effects on the cells, we are currently working on this for a follow-up detailed manuscript and it is beyond this study. However, we were able to find two recent studies that both performed gene expression and functional studies of cytokine secretion following flow-based mechanoporation, which are both highly relevant to our approach. The new discussion of these relevant studies is on **page 16, lines 298-306**.

3. Validation with Primary Cells: The absence of primary cell validation would potentially make the manuscript to the realm of conceptual exploration, devoid of tangible impact. The reviewer strongly advocates for the execution of primary CRISPR tests as an essential validation step to substantiate the device's utility and potency.

We are currently in the planning stage of a multi-center project, in collaboration with investigators in the MGH Center for Cancer Research, on gene editing in primary T cells to address these specific questions, which (we agree with the reviewer) are scientifically interesting. However, these studies are beyond the scope of this first research article which is focused on the underlying delivery technology and not a specific cell type or clinical application. This large-scale effort for CRISPR gene editing in primary T cells will need to be a separate paper.

Response to Reviewer 2

I have reviewed the author comments and the updated manuscript and feel that the current version addresses my concerns sufficiently.

We are glad that the revised manuscript has sufficiently addressed the Reviewer's concerns.

Response to Reviewer 3

NCOMMS-23-12330A is a revision of the manuscript entitled "High throughput intracellular delivery by viscoelastic mechanoporation". Overall, the authors were very responsive to the concerns raised by all reviewers and revised the manuscript appropriately to address these concerns. The manuscript was revised to include more clarity about HA and vector concentrations, volume processed, and appropriate inertio-elastic stretching times as well as including a third replicate of some experiments. In particular, the revision addresses reviewer 1's concerns about originality, performance, and cost especially in the included tables in the supplementary information file. The revision also addressed this reviewer's major concerns about delivery mechanisms, showing confocal imaging of Jurkat cells with the cell membrane dyed prior to and following viscoelastic mechanoporation and FITC-Dextran delivery. Following mechanoporation the FITC-Dextran is found to be relatively uniformly distributed throughout the cytoplasm. When examining the images in Figure S15 two things stand out to me: 1) The membrane morphology following viscoelastic mechanoporation looks different with a number of highly localized dyed puncta around the cell. It is unclear if this is due to membrane relaxation and decrease in cell volume while preserving membrane area.

We are glad the Reviewer found the revised manuscript very responsive to the concerns of all reviewers. Regarding (1), we agreed that Figure S15 showed potential changes in morphology associated with mechanoporation. To investigate this further, we performed an experiment in which we labeled the plasma membrane using a covalent dye and took images of unfixed cells both before and after viscoelastic mechanoporation using an imaging flow cytometer. We assessed three different morphological metrics provided by the ImageStream IDEAS software on these two populations. Altogether, no differences in cell size or morphology were observed. Representative images and morphology measurement results are provided in **Figure S17** and discussed in the manuscript **page 11, lines 200-204**.

2) The distribution of FITC fluorescence also shows some variation of fluorescence intensity throughout the cytoplasm with a speckled appearance and some distinct areas of higher fluorescence. It is similarly unclear if this is due to structural differences throughout the cytosol (e.g., organelle distribution). Since only 2 images are presented, no real conclusions can be

drawn, but it may be interesting to explore secondary, longer term, delivery mechanisms especially for larger vectors like plasmid DNA (MW >1M kDa).

We agree that the FITC distribution in the confocal images seemed to have a speckled appearance, and the reason for this texture was not clear from the image of just one cell. Therefore, we performed a new experiment where image texture was automatically measured on thousands of cells that had taken 70 kDa FITC-dextran by endocytosis as compared to delivery by viscoelastic mechanoporation. As an endocytosis positive control, we incubated cells for 90 minutes with a high concentration of FITC-dextran. The amount of delivered dextran was about 75-fold greater in the cells processed with viscoelastic mechanoporation. Two different metrics of FITC texture and image distribution were assessed. These measurements showed that viscoelastic mechanoporation resulted in a quantifiably smoother distribution of dextran, and quantifiably greater distribution across a larger area within the cells as compared to endocytosis. Representative images and population measurements are provided in **Figure S16** and in the manuscript **pages 10-11, lines 194-199**.

REVIEWERS' COMMENTS

Reviewer #1 (Remarks to the Author):

Three major experiments were requested for investigation previously; however, it seems that none of them were conducted where it is believed that the manuscript has been fully addressed the concerns.

Comment #1: There are flow electroporators, and one of them is now successfully commercialized (e.g., MaxCyte); therefore, it should be careful to state that the throughput of the electroporator is limited. Furthermore, it is important and recommended to show that the presented work can actually perform better rather than referencing other group's work.

Comment #2 and 3: This work is aimed to be published in a high-impact journal, and performing a stability assessment and demonstrating primary cell applicability beyond cell viability seems valid to ask.

Reviewer #3 (Remarks to the Author):

I reiterate my previous comments that the authors have been very responsive to the reviewer concerns.

This current iteration of the manuscript addresses reviewer 1's concerns about comparing mechanoporation to electroporation efficiencies and genomic functional profiling through literature review conducting such comparisons, and highlighting future work in assessing the methodology in CRISPR gene editing techniques in primary T Cells.

The revisions also addresses reviewer 2's concerns about changes in cellular morphology and localized fluorescence in the cytoplasm by analyzing morphological metrics (fluorescence intensity distribution, circularity, image intensity gradient) using the flow cytometry data and ImageStream IDEAS on confocal cell images.

These revisions have improved an already strong manuscript and I am still in support of accepting the manuscript for publication.

Response to Reviews

NCOMMS-23-12330-T

Response to Review 1

Three major experiments were requested for investigation previously; however, it seems that none of them were conducted where it is believed that the manuscript has been fully addressed the concerns.

Comment #1: There are flow electroporators, and one of them is now successfully commercialized (e.g., MaxCyte); therefore, it should be careful to state that the throughput of the electroporator is limited. Furthermore, it is important and recommended to show that the presented work can actually perform better rather than referencing other group's work.

We agree that the commercially available electroporation system from MaxCyte is relevant. Likewise, The Lonza Nucleofector LV and ThermoFisher Xenon platforms have also recently been evaluated for commercial cell therapy applications. We have included references to recent phase I clinical trials, across which all three of these platforms were evaluated to deliver CRISPR-Cas9 ribonucleoproteins for T cell engineering (**page 2, lines 46-49**). As we describe, the editing efficiencies using these platforms are uniformly lower than laboratory-scale electroporation studies targeting the same genetic locus (T cell receptor alpha constant) in similar cells.

Comment #2 and 3: This work is aimed to be published in a high-impact journal, and performing a stability assessment and demonstrating primary cell applicability beyond cell viability seems valid to ask.

We agree with the Reviewer that primary cell applicability and assessing cell stability will be valid for addressing specific clinical applications, including CAR T therapy, which the Reviewer has mentioned before. As stated previously, we are currently in the planning stage of a multi-center study which is designed to answer these important questions.

Response to Review 3

I reiterate my previous comments that the authors have been very responsive to the reviewer concerns.

This current iteration of the manuscript addresses reviewer 1's concerns about comparing mechanoporation to electroporation efficiencies and genomic functional profiling through literature review conducting such comparisons, and highlighting future work in assessing the methodology in CRISPR gene editing techniques in primary T Cells.

The revisions also addresses reviewer 2's concerns about changes in cellular morphology and localized fluorescence in the cytoplasm by analyzing morphological metrics (fluorescence

intensity distribution, circularity, image intensity gradient) using the flow cytometry data and ImageStream IDEAS on confocal cell images.

These revisions have improved an already strong manuscript and I am still in support of accepting the manuscript for publication.

We are glad that the Reviewer feels the revisions have further improved the manuscript, and wish to thank the Reviewer for their effort and consideration.